# Boosting Adversarial Transferability by Achieving Flat Local Maxima

**Zhijin Ge[1]\*, Hongying Liu[2]\*, Xiaosen Wang[3]\*, Fanhua Shang[4]†, Yuanyuan Liu[1]†**
[1]School of Artificial Intelligence, Xidian University
[2]Medical College, Tianjin University, China
[3]Huawei Singular Security Lab
[4]College of Intelligence and Computing, Tianjin University
`zhijinge@stu.xidian.edu.cn, hyliu2009@tju.edu.cn`
`xiaosen@hust.edu.cn, fhshang@tju.edu.cn, yyliu@xidian.edu.cn`

## Abstract

Transfer-based attack adopts the adversarial examples generated on the surrogate model to attack various models, making it applicable in the physical world and attracting increasing interest. Recently, various adversarial attacks have emerged to boost adversarial transferability from different perspectives. In this work, inspired by the observation that flat local minima are correlated with good generalization, we assume and empirically validate that adversarial examples at a flat local region tend to have good transferability by introducing a penalized gradient norm to the original loss function. Since directly optimizing the gradient regularization norm is computationally expensive and intractable for generating adversarial examples, we propose an approximation optimization method to simplify the gradient update of the objective function. Specifically, we randomly sample an example and adopt a first-order procedure to approximate the Hessian/vector product, which makes computing more efficient by interpolating two neighboring gradients. Meanwhile, in order to obtain a more stable gradient direction, we randomly sample multiple examples and average the gradients of these examples to reduce the variance due to random sampling during the iterative process. Extensive experimental results on the ImageNet-compatible dataset show that the proposed method can generate adversarial examples at flat local regions, and significantly improve the adversarial transferability on either normally trained models or adversarially trained models than the state-of-the-art attacks. Our codes are available at: https://github.com/Trustworthy-AI-Group/PGN.

## 1  Introduction

A great number of works have shown that Deep Neural Networks (DNNs) are vulnerable to adversarial examples [10, 16, 38, 26, 50], which are generated by applying human-imperceptible perturbations on clean input to result in misclassification. Furthermore, adversarial examples have an intriguing property of transferability [10, 11, 31, 61, 59, 57], *i.e.*, the adversarial example generated from the surrogate model can also misclass other models. The existence of transferability makes adversarial attacks practical to real-world applications because hackers do not need to know any information about the target model, which introduces a series of serious security problems in security-sensitive applications such as self-driving [65, 13] and face-recognition [44, 30].

---

*Equal Contribution.
†Corresponding authors

Although several attack methods [5, 16, 26, 28] have exhibited great attack effectiveness in the white-box setting, they have low transferability when attacking black-box models, especially for some advanced defense models [36, 47]. Previous works [10, 31, 61] attribute that the reason for adversarial examples shows weak transferability due to dropping into poor local maxima or overfitting the surrogate model, which is not likely to transfer across models. To address this issue, many methods have been proposed from different perspectives. Gradient optimization attacks [10, 31, 49, 64] attempt to boost black-box performance by advanced gradient calculation. Input transformation attacks [61, 11, 31, 51, 35, 15, 55] aim to generate adversarial examples with higher transferability by applying various transformations to the inputs. Especially, the above methods are mainly proposed from the perspective of optimization and generalization, which regard the process of generating adversarial examples on the white-box model as a standard neural network training process, and treat adversarial transferability as equivalent to model generalization [31]. Although these methods can improve the transferability of adversarial examples, there are still some gaps between white-box attacks and transfer-based black-box attacks.

Inspired by the observation that flat minima often result in better model generalization [24, 66, 41, 14], we assume and empirically validate that adversarial examples at a flat local region tend to have good transferability. Intuitively, we can achieve the flat local maxima for adversarial examples using a gradient regularization norm but it is computationally extensive to solve such a problem. To address this issue, we theoretically analyze the optimization process and propose a novel attack called Penalizing Gradient Norm (PGN). In particular, PGN approximates the Hessian/vector product by interpolating the first-order gradients of two samples. This approximation allows us to efficiently generate adversarial examples at flat local regions. To eliminate the error introduced by approximation, PGN incorporates the average gradient of several randomly sampled data points to update the adversarial perturbation. Our main contributions can be summarized as follows:

- To the best of our knowledge, it is the first work that empirically validates that adversarial examples located in flat regions have good transferability.
- We propose a novel attack called Penalizing Gradient Norm (PGN), which can effectively generate adversarial examples at flat local regions with better transferability.
- Empirical evaluations show that PGN can significantly improve the attack transferability on both normally trained models and adversarially trained models, which can also be seamlessly combined with various previous attack methods for higher transferability.

## 2 Related Work

In this section, we provide a brief overview of the adversarial attack methods and introduce several studies on the flat local minima for model generalization.

### 2.1 Adversarial Attacks

In general, adversarial attacks can be divided into two categories, *i.e.*, white-box attacks and black-box attacks. In the white-box setting, the target model is completely exposed to the attacker. For instance, Goodfellow *et al*. [16] proposed the Fast Gradient Sign Method (FGSM) to generate adversarial examples with one step of gradient update. Kurakin *et al*. [26] further extends FGSM to an iterative version with a smaller step size $\alpha$, denoted as I-FGSM. Madry *et al*. [36] extends I-FGSM with a random start to generate diverse adversarial examples. Existing white-box attacks have achieved superior performance with the knowledge of the target model. On the other hand, black-box attacks are more practical since they only access limited or no information about the target model. There are two types of black-box adversarial attacks [21]: query-based and transfer-based attacks. Query-based attacks [3, 6, 53] often take hundreds or even thousands of quires to generate adversarial examples, making them inefficient in the physical world. In contrast, transfer-based attacks [61, 49, 52, 4, 54, 56, 48] generate adversarial examples on the surrogate model, which can also attack other models without accessing the target model, leading to great practical applicability and attracting increasing attention.

Unfortunately, adversarial examples crafted by white-box attacks generally exhibit limited transferability. To boost adversarial transferability, various methods are proposed from the perspective of optimization and generalization. MI-FGSM [10] integrates momentum into I-FGSM to stabilize the update direction and escape from poor local maxima at each iteration. NI-FGSM [31] adopts

Nesterov's accelerated gradient [40] to further enhance the transferability. Wang *et al.* [49] tunned the gradient using the gradient variance of previous iteration to find a more stable gradient direction. Wang *et al.* [52] enhanced the momentum by accumulating the gradient of several data points in the direction of the previous gradient for better transferability.

Data augmentation, which has shown high effectiveness in improving model generalization [37, 63, 2, 8], has been widely studied to boost adversarial transferability. Xie *et al.* [61] adopted diverse input patterns by randomly resizing and padding to generate transferable adversarial examples. Dong *et al.* [11] utilized several translated images to optimize the adversarial perturbations, and further calculated the gradients by convolving the gradient at untranslated images with a kernel matrix for high efficiency. SIM [31] optimizes the adversarial perturbations over several scaled copies of the input images. Admix [51] mixes up a set of images randomly sampled from other categories while maintaining the original label of the input. Spectrum simulation attack (SSA) [35] transforms the input image in the frequency domain to craft more transferable adversarial examples.

Besides, some methods improve adversarial transferability from different perspectives. For instance, Liu *et al.* [32] proposed an ensemble attack, which simultaneously attacks multiple surrogate models. Wu *et al.* [59] employed an adversarial transformation network that can capture the most harmful deformations to adversarial noises. Qin *et al.* [42] injected the reverse adversarial perturbation at each step of the optimization procedure for better transferability.

## 2.2   Flat Minima

After Hochreiter *et al.* [20] pointed out that well-generalized models may have flat minima, the connection between the flatness of minima and model generalization has been studied from both empirical and theoretical perspectives [24, 41, 14, 66]. Li *et al.* [27] observed that skip connections promote flat minima, which helps explain why skip connections are necessary for training extremely deep networks. Similarly, Santurkar *et al.* [43] found that BatchNorm makes the optimization landscape significantly smooth in the training process. Sharpness-Aware Minimization (SAM) [14] improves model generalization by simultaneously minimizing the loss value and sharpness, which seeks the parameters in the neighborhoods with uniformly low loss values. Jiang *et al.* [23] studied 40 complexity measures and showed that a sharpness-based measure has the highest correlation with generalization. Zhao *et al.* [66] demonstrated that adding a gradient norm of the loss function can help the optimizer find flat local minima.

## 3   Methodology

### 3.1   Preliminaries

Given an input image $x$ with its corresponding ground-true label $y$, the deep model $f$ with parameter $\theta$ is expected to output the prediction $f(x; \theta) = y$ with high probability. Let $\mathcal{B}_\epsilon(x) = \{x' : \|x' - x\|_p \leq \epsilon\}$ be an $\epsilon$-ball of an input image $x$, where $\epsilon > 0$ is a pre-defined perturbation magnitude, and $\|\cdot\|_p$ denotes the $L_p$-norm (e.g, the $L_1$-norm). The attacker aims to find an example $x^{adv} \in \mathcal{B}_\epsilon(x)$ that misleads the classifier $f(x^{adv}; \theta) \neq y$. Let $J(x, y; \theta)$ be the loss function (*e.g.*, cross-entropy loss) of the classifier $f$. Existing white-box attacks such as [16, 26, 10] usually generate adversarial examples by solving the following maximization problem:

$$\max_{x^{adv} \in \mathcal{B}_\epsilon(x)} J(x^{adv}, y; \theta). \tag{1}$$

These attack methods mainly generate adversarial examples through gradient iterations. For instance, I-FGSM [26] iteratively updates the adversarial perturbation as follows:

$$x_{t+1}^{adv} = \Pi_{\mathcal{B}_\epsilon(x)} \left[ x_t^{adv} + \alpha \cdot \text{sign}(\nabla_{x_t^{adv}} J(x_t^{adv}, y; \theta)) \right], \quad x_0^{adv} = x, \tag{2}$$

where $\Pi_{\mathcal{B}_\epsilon(x)}(\cdot)$ projects an input into $\mathcal{B}_\epsilon(x)$, $\alpha = \epsilon/T$, and $T$ is the total number of iterations. For black-box attacks, the gradient is not accessible so that we cannot directly solve Problem (1) like I-FGSM. To address such a issue, transfer-based attacks generate adversarial examples on an accessible surrogate model, which can be transferred to fool the target models.

## 3.2 Flat Local Maxima Tend to Improve Adversarial Transferability

In general, the adversarial transferability is equivalent to model generalization if we analogize the optimization of perturbation with the model training process [31, 49]. Existing works [24, 66, 41, 14] have demonstrated that flat local minima tend to generalize better than their sharp counterparts. This inspires us to assume that adversarial examples at a flat local region w.r.t. the loss function tend to have better transferability across various models. A rigorous description is given as follows:

**Assumption 1.** *Given any small radius $\zeta > 0$ for the local region and two adversarial examples $x_1^{adv}$ and $x_2^{adv}$ for the same input image $x$, if $\max_{x' \in \mathcal{B}_\zeta(x_1^{adv})} \|\nabla_{x'} J(x', y; \theta)\|_2 < \max_{x' \in \mathcal{B}_\zeta(x_2^{adv})} \|\nabla_{x'} J(x', y; \theta)\|_2$, with high probability, $x_1^{adv}$ tends to be more transferable than $x_2^{adv}$ across various models.*

SAM [14] has indicated that a flat local minimum is an entire neighborhood having both low loss and low curvature. Zhao *et al.* [66] also demonstrated that if the loss function has a smaller gradient value, this would indicate that the loss function landscape is flatter. Here we adopt the maximum gradient in the neighborhood to evaluate the flatness of the local region. Following Assumption 1, we introduce a regularizer, which minimizes the maximum gradient in the $\zeta$-ball of the input, into Eq. (1) as follows:

$$\max_{x^{adv} \in \mathcal{B}_\epsilon(x)} \left[ J(x^{adv}, y; \theta) - \lambda \cdot \max_{x' \in \mathcal{B}_\zeta(x^{adv})} \|\nabla_{x'} J(x', y; \theta)\|_2 \right], \tag{3}$$

where $\lambda \geq 0$ is the penalty coefficient. During the optimization process, we penalize the maximum gradient to perceive the sharper regions that can not be identified by using the averaged gradients. By optimizing Problem (3), we can find that the adversarial example with a small gradient of data points in its neighborhood should be at a flat local region. However, it is impractical to calculate the maximum gradient in the $\zeta$-ball. Hence, we approximately optimize Eq. (3) by randomly sampling an example $x' \in \mathcal{B}_\zeta(x^{adv})$ at each iteration using I-FGSM [26] and MI-FGSM [10], respectively.

As shown in Fig. 1, the regularizer of gradients can significantly boost the adversarial transferability of either I-FGSM or MI-FGSM. In general, the average attack success rate is improved by a clear margin of $5.3\%$ and $7.2\%$ for I-FGSM and MI-FGSM, respectively. Such remarkable performance improvement strongly validates Assumption 1. RAP [42] also advocates that the adversarial example should be located at a flat local region. However, to the best of our knowledge, it is the first work that empirically validates that flat local maxima can result in good adversarial transferability. Based on this finding, we aim to improve the adversarial transferability by locating the adversarial examples in a flat local region, and a detailed approach for finding flat local maxima will be provided in Sec. 3.3.

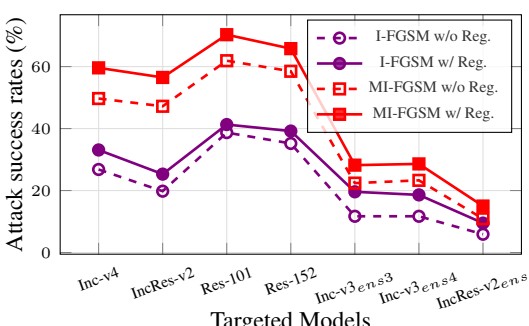

Figure 1: The average attack success rates (%) of I-FGSM and MI-FGSM w/wo the gradient regularization on seven black-box models. The adversarial examples are generated on Inc-v3.

## 3.3 Finding Flat Local Maxima

Although we have verified that adversarial examples at a flat local region tend to have better transferability, the optimization process involved in Eq. (3) is computationally expensive and intractable for generating transferable adversarial examples. The detailed ablation study is summarized in Sec. 4.6. To address this challenge, we propose an approximate optimization approach that efficiently optimizes Eq. (3) to generate more transferable adversarial examples.

Since $x'$ in Eq. (3) is close to $x^{adv}$ with a small $\zeta$, we assume that the effects of maximizing $J(x^{adv}, y; \theta)$ and maximizing $J(x', y; \theta)$ are expected to be equivalent. Then we can approximately simplify Eq. (3) as follows:

$$\max_{x^{adv} \in \mathcal{B}_\epsilon(x)} \mathcal{L}(x^{adv}, y; \theta) \approx J(x', y; \theta) - \lambda \cdot \|\nabla_{x'} J(x', y; \theta)\|_2, \quad \text{s.t.} \quad x' \in \mathcal{B}_\zeta(x^{adv}), \tag{4}$$

where $\mathcal{L}$ is a loss function. Here a penalized gradient norm is introduced into the original loss function $J$ for achieving flat local maxima.

**Algorithm 1** Penalizing Gradient Norm (PGN) attack method

---

**Input**: A clean image $x$ with ground-truth label $y$, and the loss function $J$ with parameters $\theta$.
**Parameters**: The magnitude of perturbation $\epsilon$; the maximum number of iterations, $T$; the decay factor $\mu$; the balanced coefficient $\delta$; the upper bound (i.e., $\zeta$) of random sampling in $\zeta$-ball; the number of randomly sampled examples, $N$.

1: $g_0 = 0$, $x_0^{adv} = x$, $\alpha = \epsilon/T$;
2: **for** $t = 0, 1, \cdots, T - 1$ **do**
3:     Set $\bar{g} = 0$;
4:     **for** $i = 0, 1, \cdots, N - 1$ **do**
5:         Randomly sample an example $x' \in \mathcal{B}_\zeta(x_t^{adv})$;
6:         Calculate the gradient at the sample $x'$, $g' = \nabla_{x'} J(x', y; \theta)$;
7:         Compute the predicted point by $x^* = x' - \alpha \cdot \frac{g'}{\|g'\|_1}$;
8:         Calculate the gradient of the predicted point, $g^* = \nabla_{x^*} J(x^*, y; \theta)$;
9:         Accumulate the updated gradient by $\bar{g} = \bar{g} + \frac{1}{N} \cdot [(1 - \delta) \cdot g' + \delta \cdot g^*]$;
10:    **end for**
11:    $g_{t+1} = \mu \cdot g_t + \frac{\bar{g}}{\|\bar{g}\|_1}$;
12:    Update $x_{t+1}^{adv}$ via $x_{t+1}^{adv} = \Pi_{\mathcal{B}_\epsilon(x)} \left[ x_t^{adv} + \alpha \cdot \text{sign}(g_{t+1}) \right]$;
13: **end for**
14: **return** $x^{adv} = x_T^{adv}$.

**Output**: An adversarial example $x^{adv}$.

---

In practice, it is computationally expensive to directly optimize Eq. (4), since we need to calculate its Hessian matrix. Note that existing adversarial attacks typically rely on the sign of the gradient, rather than requiring an exact gradient value. Thus, we approximate the second-order Hessian matrix by using the finite difference method to accelerate the attack process.

**Theorem 1 (Finite Difference Method [1]).** *Given a finite difference step size $\alpha$ and one normalized gradient direction vector $v = -\frac{\nabla_x J(x, y; \theta)}{\|\nabla_x J(x, y; \theta)\|_2}$, the Hessian/vector product can be approximated by the first-order gradient as follows:*

$$\nabla_x^2 J(x, y; \theta) v \approx \frac{\nabla_x J(x, y; \theta)|_{x=x+\alpha \cdot v} - \nabla_x J(x, y; \theta)}{\alpha}. \tag{5}$$

With the finite difference method, we can solve Eq. (4) approximately by using the Hessian/vector product for high efficiency. In particular, we can calculate the gradient at each iteration as follows.

**Corollary 1.** *The gradient of the objective function* (4) *at the $t$-th iteration can be approximated as:*

$$\nabla_{x_t^{adv}} \mathcal{L}(x_t^{adv}, y; \theta) \approx (1 - \delta) \cdot \nabla_{x_t'} J(x_t', y; \theta) + \delta \cdot \nabla_{x_t'} J(x_t', y; \theta)|_{x_t' = x_t' + \alpha \cdot v}, \tag{6}$$

*where $x_t'$ is a point randomly sampled in $\mathcal{B}_\zeta(x_t^{adv})$, $\alpha$ is the iteration step size, and $\delta = \frac{\lambda}{\alpha}$ is a balanced coefficient.*

The detailed proof of Corollary 1 is provided in the Appendix. From Corollary 1, we can approximate the gradient of Eq. (4) by interpolating two neighboring gradients. This approximation technique significantly enhances the efficiency of the attack process.

In Eq. (4), we approximate the loss at the input image $x^{adv}$ and the maximum gradient in $\mathcal{B}_\zeta(x^{adv})$ by randomly sampling an example $x' \in \mathcal{B}_\zeta(x^{adv})$. However, this approach introduces variance due to the random sampling process. To address this issue, we randomly sample multiple examples and average the gradients of these examples to obtain a more stable gradient. Such averaging helps reduce the error introduced by the approximation and randomness, resulting in a more accurate gradient estimation. This, in turn, allows us to effectively explore flat local maxima and achieve better transferability.

In short, we introduce a novel attack method, called Penalizing Gradient Norm (PGN). The PGN attack aims to guide adversarial examples towards flatter regions by constraining the norm or magnitude of the gradient. The details of the PGN attack are outlined in Algorithm 1. Since PGN is derived by optimizing Eq. (3) to achieve a flat local maximum, it can be seamlessly integrated with existing gradient-based attack methods and input transformation-based attack methods, leveraging their strengths to further improve adversarial transferability.

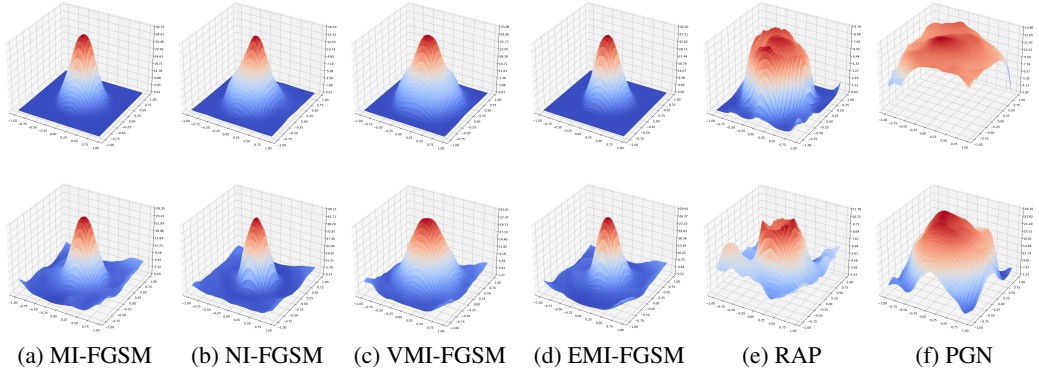

| (a) MI-FGSM | (b) NI-FGSM | (c) VMI-FGSM | (d) EMI-FGSM | (e) RAP | (f) PGN |

Figure 2: Visualization of loss surfaces along two random directions for two randomly sampled adversarial examples on the surrogate model (*i.e.*, Inc-v3). The center of each 2D graph corresponds to the adversarial example generated by different attack methods (see more examples in the Appendix).

## 4 Experiments

In this section, we conduct extensive experiments on the ImageNet-compatible dataset. We first provide the experimental setup. Then we compare the results of the proposed methods with existing methods on both normally trained models and adversarially trained models. Finally, we conduct ablation studies to study the effectiveness of key parameters in our PGN. The experimental results were performed multiple times and averaged to ensure the experimental results were reliable.

### 4.1 Experimental Settings

**Dataset.** We conduct our experiments on the ImageNet-compatible dataset, which is widely used in previous works [4, 35, 42]. It contains 1,000 images with the size of $299 \times 299 \times 3$, ground-truth labels, and target labels for targeted attacks.

**Models.** To validate the effectiveness of our methods, we test attack performance in five popular pre-trained models, including Inception-v3 (Inc-v3) [45], Inception-v4 (Inc-v4), InceptionResnet-v2 (IncRes-v2) [46], ResNet-101 (Res-101), and ResNet-152 (Res-152) [18]. We also consider adversarially trained models including, Inc-v3$_{ens3}$, Inc-v4$_{ens4}$, and IncRes-v2$_{ens}$ [47].

**Baselines.** We take five popular gradient-based iterative adversarial attacks as our baselines, including, MI-FGSM [10], NI-FGSM [31], VMI-FGSM [49], EMI-FGSM [52], RAP [42]. We also integrate the proposed method with various input transformations to validate the generality of our PGN, such as DIM [11], TIM [61], SIM [31], Admix [51], and SSA [35].

**Hyper-parameters.** We set the maximum perturbation of the parameter $\epsilon = 16.0/255$, the number of iterations $T = 10$, and the step size $\alpha = \epsilon/T$. For MI-FGSM and NI-FGSM, we set the decay factor $\mu = 1.0$. For VMI-FGSM, we set the number of sampled examples $N = 20$ and the upper bound of neighborhood size $\beta = 1.5 \times \epsilon$. For EMI-FGSM, we set the number of examples $N = 11$, the sampling interval bound $\eta = 7$, and adopt the linear sampling. For the attack method, RAP, we set the step size $\alpha = 2.0/255$, the number of iterations $K = 400$, the inner iteration number $T = 10$, the late-start $K_{LS} = 100$, the size of neighborhoods $\epsilon_n = 16.0/255$. For our proposed PGN, we set the number of examples $N = 20$, the balanced coefficient $\delta = 0.5$, and the upper bound of $\zeta = 3.0 \times \epsilon$.

### 4.2 Visualization of Loss Surfaces for Adversarial Example

To validate that our proposed PGN method can help the adversarial examples find a flat maxima region, we compare the loss surface maps of the adversarial examples generated by different attack methods on the surrogate model (*i.e.*, Inc-v3). Each 2D graph corresponds to an adversarial example, with the adversarial example shown at the center. We randomly select two images from the dataset and compare the loss surfaces in Fig. 2, each row represents the visualization of one image. From the comparison, we observe that our PGN method can help adversarial examples achieve flatter maxima regions compared to the baselines. The adversarial examples generated by our method are located

Table 1: The untargeted attack success rates (%±std, over 10 random runs) of various gradient-based attacks in the single model setting. The adversarial examples are crafted on Inc-v3, Inc-v4, IncRes-v2, and Res-101 by MI-FGSM (MI), NI-FGSM (NI), VMI-FGSM (VMI), EMI-FGSM (EMI), RAP, and our PGN attack methods, respectively. Here * indicates the white-box model.

| Model | Attack | Inc-v3 | Inc-v4 | IncRes-v2 | Res-101 | Inc-v3$_{ens3}$ | Inc-v3$_{ens4}$ | IncRes-v2$_{ens}$ |
|---|---|---|---|---|---|---|---|---|
| | MI | **100.0±0.0*** | 51.0±0.47 | 45.8±0.60 | 49.0±0.24 | 22.6±0.52 | 22.0±0.35 | 10.9±0.24 |
| | NI | **100.0±0.0*** | 61.4±0.42 | 59.6±0.54 | 57.2±0.18 | 22.5±0.37 | 22.7±0.35 | 11.5±0.26 |
| | VMI | **100.0±0.0*** | 74.8±0.58 | 69.9±0.92 | 65.5±0.67 | 41.6±0.54 | 41.6±0.54 | 25.0±0.34 |
| Inc-v3 | EMI | **100.0±0.0*** | 80.7±0.58 | 77.1±0.37 | 72.4±0.83 | 33.0±0.61 | 31.9±0.49 | 17.0±0.48 |
| | RAP | 99.9±0.10* | 84.5±0.69 | 79.3±0.47 | 76.5±0.65 | 56.9±0.84 | 51.3±0.62 | 31.9±0.35 |
| | PGN | **100.0±0.0*** | **90.6±0.67** | **89.5±0.75** | **81.2±0.68** | **64.6±0.75** | **65.6±0.94** | **45.3±0.77** |
| | MI | 57.2±0.36 | **100.0±0.0*** | 46.1±0.14 | 51.5±0.33 | 19.1±0.46 | 18.4±0.23 | 10.2±0.36 |
| | NI | 62.8±0.43 | **100.0±0.0*** | 52.7±0.34 | 56.7±0.19 | 19.2±0.25 | 18.3±0.37 | 11.7±0.29 |
| | VMI | 77.6±0.65 | 99.8±0.10* | 69.8±0.41 | 66.7±0.33 | 41.1±0.87 | 41.2±0.54 | 27.0±0.24 |
| Inc-v4 | EMI | 84.2±0.62 | 99.7±0.10* | 75.0±0.70 | 74.4±0.64 | 31.5±0.44 | 28.0±0.65 | 16.2±0.36 |
| | RAP | 85.3±0.74 | 99.5±0.21* | 79.5±0.62 | 77.2±0.42 | 45.2±0.69 | 46.8±0.48 | 29.3±0.51 |
| | PGN | **91.2±0.58** | 99.6±0.15* | **87.6±0.74** | **83.5±0.53** | **67.0±0.68** | **64.2±0.63** | **49.1±0.82** |
| | MI | 58.2±0.21 | 52.4±0.41 | 99.3±0.21* | 50.7±0.26 | 22.0±0.37 | 22.0±0.31 | 13.8±0.43 |
| | NI | 60.3±0.35 | 57.1±0.17 | 99.5±0.17* | 55.3±0.35 | 18.3±0.18 | 19.3±0.29 | 12.1±0.16 |
| | VMI | 78.2±0.64 | 77.0±0.57 | 99.1±0.36* | 66.0±0.48 | 47.6±0.69 | 43.3±0.36 | 37.7±0.37 |
| IncRes-v2 | EMI | 85.2±0.78 | 83.3±0.29 | 99.7±0.18* | 74.0±0.56 | 38.4±0.48 | 33.8±0.53 | 24.1±0.48 |
| | RAP | 87.1±0.75 | 84.2±0.45 | 99.4±0.28* | 79.4±0.64 | 50.3±0.47 | 49.8±0.89 | 40.2±0.54 |
| | PGN | **92.0±0.69** | **92.3±0.63** | **99.8±0.10*** | **83.5±0.41** | **74.6±0.75** | **71.5±0.64** | **66.62±0.58** |
| | MI | 51.5±0.26 | 42.2±0.35 | 36.3±0.24 | **100.0±0.0*** | 18.7±0.32 | 16.6±0.14 | 9.0±0.22 |
| | NI | 55.6±0.35 | 46.9±0.41 | 40.8±0.28 | **100.0±0.0*** | 17.5±0.57 | 17.6±0.42 | 9.2±0.24 |
| | VMI | 75.0±0.40 | 69.2±0.59 | 63.0±0.84 | **100.0±0.0*** | 35.9±0.41 | 35.7±0.87 | 24.1±0.57 |
| Res-101 | EMI | 74.3±0.65 | 71.7±0.47 | 62.6±0.29 | **100.0±0.0*** | 25.7±0.74 | 24.6±0.98 | 13.3±0.68 |
| | RAP | 80.4±0.75 | 75.5±0.56 | 68.0±0.84 | 99.9±0.10* | 40.3±0.47 | 39.9±0.73 | 30.4±1.03 |
| | PGN | **86.2±0.84** | **83.3±0.66** | **77.8±0.69** | **100.0±0.0*** | **63.1±1.32** | **62.9±0.74** | **50.8±0.88** |

in larger and smoother flat local maxima. This confirms that our method can generate adversarial examples located in the flat maximum. With the adversarial examples located in more flat local maxima, PGN exhibits much better transferability than existing attacks as shown in Sec. 4.3- 4.5, which further supports our motivation.

## 4.3 Attack a Single Model

We conduct a series of gradient-based attacks under a single-model setting and report the attack success rates, which indicate the misclassification rates of the target models when using adversarial examples as inputs. The adversarial examples are generated on four different models: Inc-v3, Inc-v4, IncRes-v2, and Res-101, respectively. The results are summarized in Table 1.

From the results, we observe that our PGN method not only maintains a high attack success rate on white-box models but also significantly improves the attack success rate on black-box models. For example, when generating adversarial examples on Inc-v3, VMI-FGSM, EMI-FGSM, RAP achieve the attack success rates of 74.8%, 80.7% and 84.5%, respectively on Inc-v4. In comparison, our PGN method achieves an impressive attack success rate of 90.6%, surpassing RAP (the best baseline) by a margin of 6.1%. Moreover, when targeting adversarially trained models, our PGN attack method consistently outperforms other gradient-based attacks, improving the attack success rate by at least 11.8% compared to state-of-the-art methods on average. This convincingly validates the high effectiveness of our proposed method against both normally trained and adversarially trained models. Such outstanding results highlight the effectiveness of locating adversarial examples in flat regions for improved transferability, which is consistent with our motivation.

## 4.4 Attack an Ensemble of Models

In addition to attacking a single model, we also evaluate the performance of our PGN method in an ensemble-model setting to further validate its effectiveness. In this subsection, we adopt the ensemble attack method in [10], which creates an ensemble by averaging the logit outputs of different models. Specifically, the adversaries are generated by integrating three normally trained models, including

Table 2: The untargeted attack success rates (%) of various gradient-based attacks on eight models in the multi-model setting. The adversarial examples are generated on the ensemble models, *i.e.* Inc-v3, Inc-v4, and IncRes-v2. Here * indicates the white-box model.

| Attack | Inc-v3 | Inc-v4 | IncRes-v2 | Res-101 | Res-152 | Inc-v3$_{ens3}$ | Inc-v3$_{ens4}$ | IncRes-v2$_{ens}$ | Avg. |
|--------|--------|--------|-----------|---------|---------|-----------------|-----------------|-------------------|------|
| MI | 99.8* | 99.5* | 97.8* | 66.8 | 68.4 | 36.7 | 36.9 | 22.6 | 66.06 |
| NI | **100.0*** | **99.9*** | 99.6* | 74.6 | 74.0 | 37.7 | 37.1 | 22.2 | 68.14 |
| VMI | 99.9* | 99.5* | 98.0* | 82.2 | 81.6 | 66.8 | 64.1 | 50.8 | 80.36 |
| EMI | **100.0*** | **99.9*** | **99.7*** | 92.3 | 93.0 | 62.8 | 59.5 | 40.0 | 80.90 |
| RAP | **100.0*** | 99.4* | 98.2* | 93.5 | 93.4 | 70.1 | 69.8 | 58.9 | 85.41 |
| PGN | **100.0*** | **99.9*** | 99.6* | **94.2** | **94.6** | **88.2** | **86.6** | **79.2** | **92.78** |

Table 3: The untargeted attack success rates (%) of our PGN method, when it is integrated with DIM, TIM, SIM, Admix, and SSA, respectively. The adversarial examples are generated on Inc-v3. Here * indicates the white-box model.

| Attack | Inc-v3 | Inc-v4 | IncRes-v2 | Res-101 | Inc-v3$_{ens3}$ | Inc-v3$_{ens4}$ | IncRes-v2$_{ens}$ | Avg. |
|--------|--------|--------|-----------|---------|-----------------|-----------------|-------------------|------|
| DIM | 99.7* | 72.2 | 67.3 | 62.8 | 32.8 | 30.7 | 16.4 | 54.56 |
| PGN-DIM | **100.0*** | **93.6** | **91.9** | **87.3** | **78.3** | **77.5** | **59.8** | **84.06** |
| TIM | 99.9* | 51.6 | 47.2 | 47.8 | 29.6 | 30.7 | 20.5 | 46.76 |
| PGN-TIM | **100.0*** | **87.6** | **84.1** | **75.0** | **78.1** | **77.6** | **65.5** | **81.13** |
| SIM | **100.0*** | 70.5 | 68.2 | 63.8 | 37.5 | 37.8 | 22.0 | 57.11 |
| PGN-SIM | **100.0*** | **92.5** | **91.2** | **84.0** | **76.1** | **75.7** | **59.0** | **82.64** |
| Admix | **100.0*** | 78.6 | 77.3 | 69.5 | 41.6 | 40.3 | 24.1 | 61.63 |
| PGN-Admix | **100.0*** | **93.1** | **92.2** | **85.5** | **76.9** | **77.2** | **60.2** | **83.57** |
| SSA | 99.7* | 88.3 | 86.8 | 77.7 | 56.7 | 55.3 | 35.2 | 71.39 |
| PGN-SSA | **99.8*** | **89.9** | **89.7** | **82.9** | **69.2** | **67.8** | **47.1** | **78.03** |

Inc-v3, Inc-v4, and IncRes-v2. All the ensemble models are assigned equal weights and we test the performance of transferability on both normally trained models and adversarially trained models.

The results, presented in Table 2, demonstrate that our PGN method consistently achieves the highest attack success rates in the black-box setting. Compared to previous gradient-based attack methods, PGN achieves an average success rate of 92.78%, outperforming VMI-FGSM, EMI-FGSM, and RAP by 12.42%, 11.88%, and 7.37%, respectively. Notably, our method exhibits even greater improvements against adversarially trained models, surpassing the best attack method RAP by over 18.4% on average. These results validate that incorporating penalized gradient norms into the loss function effectively enhances the transferability of adversarial attacks, which also confirms the superiority of our proposed method in adversarial attacks.

## 4.5 Combined with Input Transformation Attacks

Existing input transformation-based attacks have shown great compatibility with each other. Similarly, due to the simple and efficient gradient update process, our proposed PGN method can also be combined with these input transformation-based methods to improve the transferability of adversarial examples. To further demonstrate the efficacy of the proposed PGN method, we integrate our method into these input transformations *i.e.*, DIM, TIM, SIM, Admix, and SSA. We generate adversarial examples on the Inc-v3 model and test the transferability of adversarial examples on six black-box models.

The experimental results are shown in Table 3. When combined with our gradient update strategy, it can significantly improve the adversarial transferability of these input transformation-based attack methods in the black-box setting. At the same time, our method also has great improvement after combining these methods. For example, DIM only achieves an average success rate of 54.56% on the seven models, while when combined with our PGN method it can achieve an average rate of 84.06%, which is 29.5% higher than before. Especially, after combining these input transformation-based methods, our PGN tends to achieve much better results on the ensemble adversarially trained models, compared with the results in Table 1. It is a significant improvement and shows that our method

Table 4: Comparison of the approximation effect between directly optimizing the Hessian matrix ($H_m$) and using the Finite Difference Method (FDM) to approximate. "Time" represents the total running time on 1,000 images, and "Memory" represents the computing memory size.

| Attack | $H_m$ | FDM | Inc-v3 | Inc-v4 | IncRes-v2 | Res-101 | Res-152 | Time (s) | Memory (MiB) |
|---|---|---|---|---|---|---|---|---|---|
| | ✗ | ✗ | 100.0* | 27.8 | 19.1 | 38.1 | 35.2 | 52.31 | 1631 |
| I-FGSM | ✓ | ✗ | 100.0* | **39.2** | **30.2** | **47.0** | **45.5** | **469.54** | **7887** |
| | ✓ | ✓ | 100.0* | 37.9 | 28.6 | 45.7 | 44.6 | 96.42 | 1631 |

has good scalability and can be combined with existing methods to further improve the success rate of transfer-based black-box attacks. In addition, our PGN method can also be combined with various gradient-based attack methods to enhance the transferability of previous works. The more experimental results are shown in the Appendix.

## 4.6 Ablation Study on Finite Difference Method

In this subsection, we will analyze and experimentally verify the effectiveness of the finite difference method in accelerating the approximation of the second-order Hessian matrix. We first theoretically analyze the acceleration effect of the finite difference method and substantiate our theoretical analysis with comparative experiments.

**Theoretical analysis.** For the baseline attack method, I-FGSM [26], the gradient is computed only once per iteration. Thus, its computational complexity is $O(n)$, where $n$ represents the image size. However, when we introduce the penalty gradient term, the need arises to compute the second-order Hessian matrix, leading to a theoretical computational complexity of $O(n^2)$. To address this, we propose the finite difference method as an approximation to the Hessian matrix, which requires the computation of the gradient twice in each iteration, effectively yielding a computational complexity of $O(2n)$. This theoretically promises significant improvements in computational efficiency.

**Experimental comparison.** To validate the effectiveness of the finite difference method in accelerating computation, we compare the runtime and computational memory before and after using the finite-difference method. These experiments were conducted using codes executed on an RTX 2080 Ti with a CUDA environment. We employed I-FGSM and evaluated the total running time on 1,000 images (excluding data loading time) and the attack success rate on black-box models. The outcomes are presented in Table 4. Directly optimizing Eq. (4) results in better attack performance with high computational resources. With the finite difference method (FDM), we can better approximate the performance of direct optimization of the second-order Hessian matrix, which significantly reduces the running time and the computational memory. Furthermore, owing to the relatively modest image size and the comparatively small number of parameters compared to the model, the accelerated computing capabilities of CUDA enable the actual running time to be less than the theoretical estimates.

## 4.7 Ablation Study on Hyper-parameters

In this subsection, we conduct a series of ablation experiments to study the impact of different parameters, including the balanced coefficient $\delta$ and the upper bound of $\zeta$-ball, and the study of the number of sampled examples $N$ will be illustrated in the Appendix. To simplify our analysis, we only consider the transferability of adversarial examples crafted on the Inc-v3 model.

**The balanced coefficient $\delta$.** In Sec. 3.3, we introduce a balanced coefficient $\delta$ to represent the penalty coefficient $\lambda$ (see the Appendix for details). Compared to studying the parameter of $\lambda$, studying $\delta$ will be much easier because it divides the scope of the parameter learning. As shown in Fig. 3a, we study the influence of the $\delta$ in the black-box setting where $\zeta$ is fixed to $3.0 \times \epsilon$. As we increase $\delta$, the transferability increases and achieves the peak for these black-box models when $\delta = 0.5$. This indicates that when averaging these two gradients the performance is best. Therefore, we set $\delta = 0.5$ in our experiments.

**The upper bound of $\zeta$-ball.** In Fig. 3b, we study the influence of the upper bound neighborhood size for random sampling, determined by parameter $\zeta$, on the success rates in the black-box setting. In our experiments, we use uniform sampling, which reduces the bias due to uneven sample distribution. As

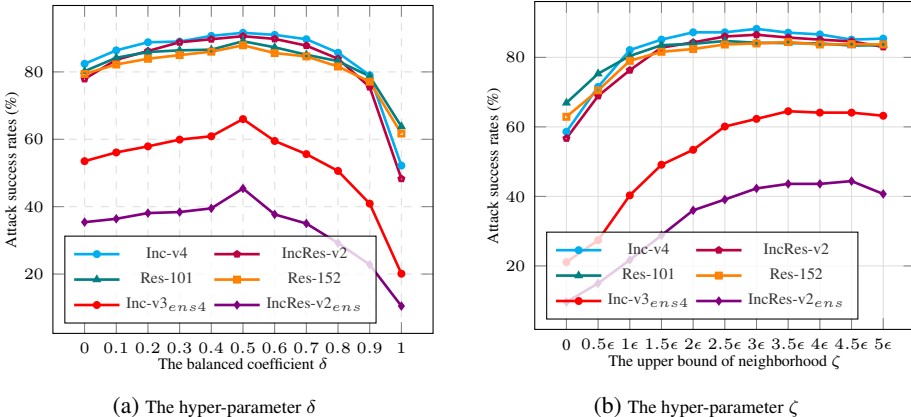

(a) The hyper-parameter $\delta$        (b) The hyper-parameter $\zeta$

Figure 3: The untargeted attack success rates (%) on six black-box models with various hyper-parameters $\delta$ or $\zeta$. The adversarial examples are generated by PGN on Inc-v3.

we increase $\zeta$, the transferability increases and achieves the peak for normally trained models when $\zeta = 3.0 \times \epsilon$, but it is still increasing against the adversarially trained models. When $\zeta > 4.5 \times \epsilon$, the performance of adversarial transferability will decrease on seven black-box models. To achieve the trade-off for the transferability on normally trained models and adversarially trained models, we set $\zeta = 3.0 \times \epsilon$ in our experiments.

## 5 Conclusion

Inspired by the observation that flat local minima often result in better generalization, we hypothesize and empirically validate that adversarial examples at a flat local region tend to have better adversarial transferability. Intuitively, we can optimize the perturbation with a gradient regularize in the neighborhood of the input sample to generate an adversarial example in a flat local region but it is non-trivial to solve such an objective function. To address such an issue, we propose a novel attack method called Penalizing Gradient Norm (PGN). Specifically, PGN approximates the Hessian/vector product by interpolating the first-order gradients of two samples. To better explore its neighborhood, PGN adopts the average gradient of several randomly sampled data points to update the adversarial perturbation. Extensive experiments on the ImageNet-compatible dataset demonstrate that PGN can generate adversarial examples at more flat local regions and achieve much better transferability than existing transfer-based attacks. Our PGN can be seamlessly integrated with other gradient-based and input transformation-based attacks to further improve adversarial transferability, demonstrating its versatility and ability to improve adversarial transferability across different attack scenarios.

## 6 Limitation

Although we have experimentally verified that flat local minima can improve the transferability of adversarial attacks, there is still a lack of theoretical analysis regarding the relationship between flatness and transferability. An existing perspective is that transferability may be related to the generalization ability of flatness. We will also keep studying the theoretical connection between transferability and flat local minima in our future work. We hope our work sheds light on the potential of flat local maxima in generating transferable adversarial examples and provides valuable insights for further exploration in the field of adversarial attacks.

## Acknowledgments

We want to thank the anonymous reviewers for their valuable suggestions and comments. This work was supported by the National Natural Science Foundation of China (Nos. 62276182, 62072334, and 61976164), and the National Science Basic Research Plan in Shaanxi Province of China (No. 2022GY-061).

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

# A Proof of Corollary 1

In this work, we introduce a penalized gradient norm to the original loss function, which helps the adversarial examples to achieve a flat maximum. Then we randomly sample an example $x'$ in the neighborhood of the adversarial example $x^{adv}$ and simplify the objective function as follows:

$$\max_{x^{adv} \in \mathcal{B}_\epsilon(x)} \mathcal{L}(x^{adv}, y; \theta) \approx [J(x', y; \theta) - \lambda \cdot \|\nabla_{x'} J(x', y; \theta)\|_2], \quad \text{s.t.} \quad x' \in \mathcal{B}_\zeta(x^{adv}). \quad (7)$$

Gradient-based attacks require calculating the gradient of the objective function during practical optimization, thus the gradient of the current loss function (7) can be expressed as follows:

$$\nabla_{x^{adv}} \mathcal{L}(x^{adv}, y; \theta) \approx \nabla_{x'} J(x', y; \theta) - \lambda \cdot \nabla_{x'}(\|\nabla_{x'} J(x', y; \theta)\|_2)$$
$$\approx \nabla_{x'} J(x', y; \theta) - \lambda \cdot \nabla_{x'}^2 J(x', y; \theta) \cdot \frac{\nabla_{x'} J(x', y; \theta)}{\|\nabla_{x'} J(x', y; \theta)\|_2}. \quad (8)$$

In practice, it is computationally expensive to directly optimize Eq. (8), since we need to calculate the Hessian matrix. In this work, we approximate the second-order Hessian matrix using the finite difference method to accelerate the attack process. Specifically, local Taylor expansion would be employed to approximate the operation results between the Hessian matrix and the gradient vector.

## A.1 Proof of Theorem 1

*Proof.* According to the Taylor expansion, we have

$$\nabla_x J(x + \Delta x, y; \theta) = \nabla_x J(x, y; \theta) + \nabla_x^2 J(x, y; \theta)\Delta x + O(\|\Delta x\|^2), \quad (9)$$

where $\Delta x = \alpha \cdot v$, $\alpha$ is a small step size, and $v$ is a normalized gradient direction vector. Here, we denote $v = -\frac{\nabla_x J(x, y; \theta)}{\|\nabla_x J(x, y; \theta)\|_2}$.

Therefore, the Hessian/vector product can be approximated by the first-order gradient as follows:

$$\nabla_x^2 J(x, y; \theta)v \approx \frac{\nabla_x J(x + \alpha \cdot v, y; \theta) - \nabla_x J(x, y; \theta)}{\alpha}. \quad (10)$$

$\square$

## A.2 Proof of Corollary 1

*Proof.* From Eqs. (8) and (10), the gradient of the loss function $\mathcal{L}(\cdot)$ can be expressed as:

$$\nabla_{x^{adv}} \mathcal{L}(x^{adv}, y; \theta) \approx \nabla_{x'} J(x', y; \theta) - \lambda \cdot \nabla_{x'}^2 J(x', y; \theta) \cdot \frac{\nabla_{x'} J(x', y; \theta)}{\|\nabla_{x'} J(x', y; \theta)\|_2}$$
$$\approx \nabla_{x'} J(x', y; \theta) + \lambda \cdot \frac{\nabla_{x'} J(x' + \alpha \cdot v, y; \theta) - \nabla_{x'} J(x', y; \theta)}{\alpha} \quad (11)$$
$$= (1 - \frac{\lambda}{\alpha}) \cdot \nabla_{x'} J(x', y; \theta) + \frac{\lambda}{\alpha} \cdot \nabla_{x'} J(x' + \alpha \cdot v, y; \theta).$$

We introduce a balanced coefficient $\delta$ and denote it as $\delta = \frac{\lambda}{\alpha}$. Hence, the gradient of the objective function (7) at the $t$-th iteration can be approximated as:

$$\nabla_{x_t^{adv}} \mathcal{L}(x_t^{adv}, y; \theta) \approx (1 - \delta) \cdot \nabla_{x_t'} J(x_t', y; \theta) + \delta \cdot \nabla_{x_t'} J(x_t' + \alpha \cdot v, y; \theta). \quad (12)$$

$\square$

# B Visualization of Loss Surfaces

**Implementation details.** Given that the adversarial example $x^{adv}$ typically has a large number of dimensions, visualizing the loss function against all dimensions becomes infeasible. To this end, we randomly select two directions, denoted as $r_1$ and $r_2$, from a Gaussian distribution with the same dimension as $x^{adv}$. Next, we calculate the loss change by varying the magnitudes of

$k_1$ and $k_2$, representing the scaling factors applied to $r_1$ and $r_2$, respectively, which enables us to visualize the loss function using a two-dimensional plot. This approach provides a slice of the loss function, allowing us to analyze its behavior and understand the impact of perturbations along different directions.

**Visualization of loss surfaces for more adversarial examples.** We visualize five randomly selected images in the ImageNet-compatible dataset. The adversarial examples are generated by various gradient-based attack methods on Inc-v3. These selected images include the three images that can be successfully transferred using our PGN method but cannot be transferred using other baseline methods. As shown in Fig. 4, we can observe that our method generates visually similar adversaries as other attacks. Hence, our method demonstrates the capability to guide adversarial examples towards larger and smoother flat regions. This observation substantiates the effectiveness of our PGN method in generating adversarial examples that reside within flat regions, thereby shedding light on the potential role of flat local maxima in generating transferable adversarial examples.

# C Combined with Other Categories of Attacks

## C.1 Gradient-based Attacks

Our PGN attack method can also be combined with various gradient-based attacks. The core of our method involves updating gradients by interpolating the first-order gradients from two samples to approximately minimize the gradient norm. In contrast, conventional gradient-based methods typically utilize a single example for gradient updates. To evaluate the efficacy of our strategy, we incorporate this interpolation approach into previous gradient-based methods, such as I-FGSM (BIM), MI-FGSM, NI-FGSM, VMI-FGSM, EMI-FGSM, and RAP. To simplify the experimental setup, we omitted random sampling and directly substituted the gradient update process of these methods with our proposed strategy.

The experimental results are presented in Table 5. Notably, when our gradient update strategy is integrated, there is a remarkable improvement in the adversarial transferability of the gradient-based attack methods in the black-box setting. For example, RAP alone achieves an average success rate of $68.80\%$ across the seven models. However, when combined with our PGN method, the average success rate rises to $73.91\%$, exhibiting a significant improvement of $5.11\%$. This outcome underscores the robust scalability of our approach, as it seamlessly integrates with existing methodologies to further amplify the success rate of transfer-based attacks.

Table 5: Untargeted attack success rates (%) of our PGN method, when it is integrated with I-FGSM (BIM), MI-FGSM, NI-FGSM, VMI-FGSM, EMI-FGSM, and RAP, respectively. The adversarial examples are generated on Inc-v3. * indicates the white-box model.

| Attack | Inc-v3 | Inc-v4 | IncRes-v2 | Res-101 | Inc-v3$_{ens3}$ | Inc-v3$_{ens4}$ | IncRes-v2$_{ens}$ | Avg. |
|---|---|---|---|---|---|---|---|---|
| BIM | **100.0*** | 28.1 | 20.6 | 26.7 | 11.9 | 11.9 | 5.0 | 29.17 |
| PGN-BIM | **100.0*** | **33.1** | **25.3** | **30.4** | **13.7** | **13.8** | **6.1** | **31.77** |
| MI | **100.0*** | 50.8 | 46.3 | 48.9 | 23.3 | 22.2 | 11.7 | 43.31 |
| PGN-MI | **100.0*** | **56.4** | **53.3** | **55.0** | **24.7** | **24.7** | **11.8** | **46.54** |
| NI | **100.0*** | 62.3 | 59.8 | 57.8 | 22.4 | 22.3 | 11.8 | 48.06 |
| PGN-NI | **100.0*** | **68.6** | **65.8** | **61.9** | **25.7** | **25.8** | **12.8** | **51.51** |
| VMI | **100.0*** | 75.2 | 70.2 | 66.0 | 41.7 | 40.9 | 24.8 | 59.83 |
| PGN-VMI | **100.0*** | **79.8** | **75.9** | **69.8** | **45.1** | **46.1** | **27.8** | **63.50** |
| EMI | **100.0*** | 81.6 | 77.0 | 72.0 | 32.8 | 32.2 | 17.5 | 59.01 |
| PGN-EMI | **100.0*** | **83.6** | **81.7** | **75.9** | **35.2** | **34.4** | **17.5** | **61.19** |
| RAP | 99.9 | 84.6 | 79.3 | 76.6 | 57.2 | 51.3 | 32.7 | 68.80 |
| PGN-RAP | **100.0*** | **90.3** | **86.4** | **85.7** | **59.5** | **56.3** | **39.3** | **73.91** |

## C.2 Skip Gradient Method

Moreover, our method can also be combined with the Skip Gradient Method (SGM) [58]. By using MI-FGSM as the backbone method, we first compare our PGN method with SGM on the

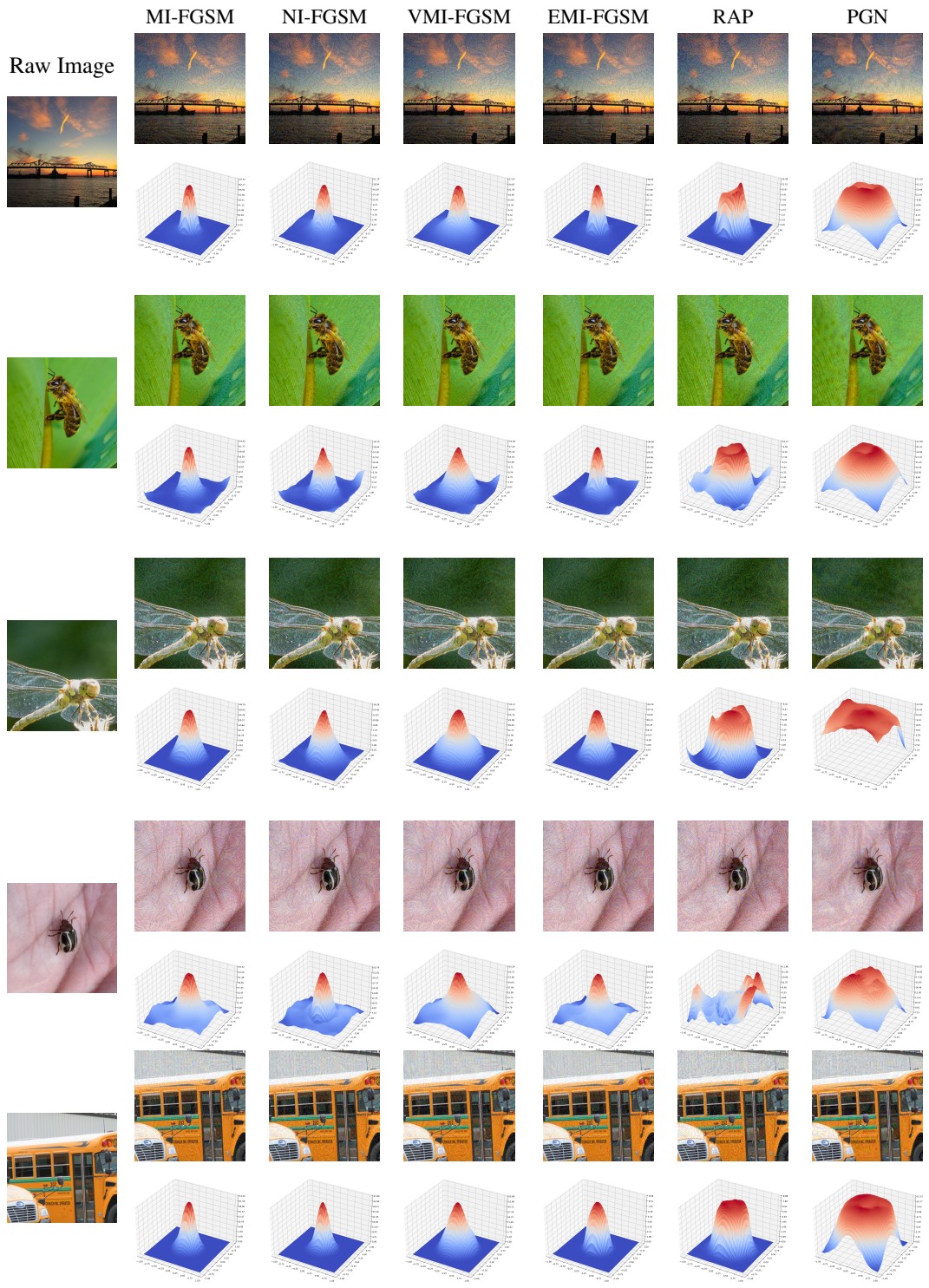

Figure 4: Visualization of adversarial examples with their corresponding loss surfaces along two random directions. Here, we randomly sampled five images and generated the adversarial examples on Inc-v3. These selected images include the three images (i.e., the last three ones) that can be successfully transferred using our PGN method but cannot be transferred using other baseline methods. The loss surfaces are also calculated on Inc-v3.

ResNet-18 model. Since SGM modifies the backpropagation and is compatible with our PGN, we also integrate PGN into SGM to evaluate its generality to other attacks. As shown in Table 6, our PGN consistently exhibits better attack performance than SGM, which further shows its superiority in boosting adversarial transferability. Besides, PGN-SGM outperforms both PGN and SGM, showing its remarkable compatibility with various attacks.

Table 6: Untargeted attack success rates (%) of our PGN method with SGM method in the single model setting. The adversarial examples are crafted on Res-18.

| Source:Res-18 | Res-101 | Res-152 | Inc-v3 | Inc-v4 | IncRes-v2 | Inc-v3$_{ens3}$ | Inc-v3$_{ens4}$ | Avg. |
|---|---|---|---|---|---|---|---|---|
| MI | 82.8 | 73.3 | 54.5 | 48.7 | 33.9 | 17.4 | 18.1 | 46.96 |
| SGM | 89.1 | 82.5 | 66.0 | 58.8 | 45.4 | 20.8 | 20.8 | 54.77 |
| PGN | **95.9** | **92.9** | **77.9** | **75.9** | **58.7** | **33.7** | **36.3** | **67.33** |
| PGN-SGM | **96.7** | **93.8** | **80.1** | **76.3** | **62.8** | **33.8** | **37.0** | **68.64** |

# D  Ablation Studies for More Hyper-parameters

## D.1  The Number of Sampled Examples, $N$

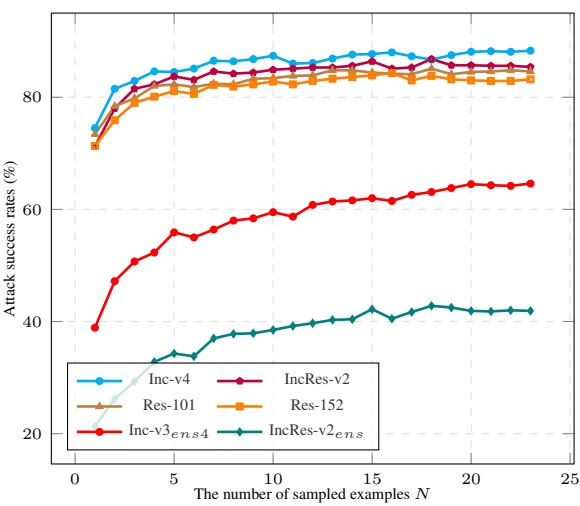

Figure 5: Untargeted attack success rates (%) on six black-box models with the different number of sampled samples $N$. The adversarial examples are generated by PGN on Inc-v3.

In this study, we employ random sampling of multiple examples and calculate the average gradients of these examples to mitigate the variance resulting from random sampling during the iterative process. To investigate the influence of the number of sampled examples, denoted as $N$, we conduct ablation experiments to analyze this parameter. As illustrated in Figure 5, when $N = 1$, our method demonstrates the lowest level of transferability. However, as we increase the value of $N$, the transferability exhibits rapid improvement until $N = 12$, after which it gradually converges for normally trained models. Notably, when $N > 12$, a slight performance improvement can still be achieved by increasing the number of sampled examples in our PGN method. To strike a balance between transferability and computational overhead, we set $N = 20$ in our work. This observation further substantiates that sampling random examples from the vicinity of the adversarial example effectively facilitates neighborhood exploration. Consequently, it stabilizes the gradient update process and encourages the discovery of flatter regions by the adversarial example.

## D.2  Finite Difference Step-size $\alpha$

In this work, we simply set the finite difference step-size to $\alpha = 1.6/255 \approx 6.27\text{e-}3$ to avoid redundant hyperparameters. In order to study the effect of the smaller step-size $\alpha$ on the finite

difference method, we performed more experiments. We investigate the transferability of adversarial examples when the alpha is set to 1e-4 and 1e-5, and the experimental results are reported in Table 7. The results show that smaller step-sizes cannot improve adversarial transferability. This is because the smaller the step-size, the closer the two neighboring gradients are, leading our method to degrade to MI-FGSM.

With the current choice of $\alpha$, our method also approximates the actual Hessian/vector product. We compared the cosine similarity of perturbations generated by the Hessian/vector product and the finite difference method. The average cosine similarity of 1,000 images is about 0.9035. Note that existing adversarial attacks typically rely on the sign of the gradient, rather than requiring an exact gradient value. Thus, we can approximate the second-order Hessian matrix by using the finite difference method to accelerate the attack process.

Table 7: Untargeted attack success rate (%) when using smaller step sizes in the finite-difference method. * indicates the white-box model.

| Size | Inc-v3 | Inc-v4 | IncRes-v2 | Res-101 |
|---|---|---|---|---|
| $6.27e-3$ | 100.0* | 91.6 | 90.6 | 89.1 |
| $1e-4$ | 100.0* | 86.4 | 84.6 | 82.6 |
| $1e-5$ | 100.0* | 85.3 | 85.0 | 82.4 |

# E    Differences between Our method and Related Works

There are two previous works similar to our proposed attack method (i.e., PGN), where one is Reverse Adversarial Perturbation (RAP) [42], which is an adversarial attack method. And the other is Gradient Norm Penalty (GNP) [66], which was proposed to improve generalization. To demonstrate the novelty and significance of the proposed method, we give a detailed discussion about the differences and similarities compared to these two works.

## E.1    Differences with RAP

Reverse adversarial perturbation (RAP) is a closely related work that encourages adversarial examples to be located at a region with low loss values. However, there are still many differences between our method and RAP.

**Empirical verification.** Both RAP and our PGN aim to achieve flat local optima for more transferable adversarial examples. Moreover,this is the first work to provide empirical verification that adversarial examples in flat local optima are more transferable.

**Methodology.** RAP injects worst-case perturbations into the optimization process to maximize the loss and achieve flat local optima. In contrast, PGN introduces a gradient norm into the loss to achieve a flat local optimum. Hence, the proposed method differs significantly from RAP.

**Efficiency.** Our method is more computationally efficient than RAP. In RAP, the outer loop is $400$, and the inner loop is $8$. In contrast, our PGN method employs finite differences to approximate the second-order Hessian matrix and samples multiple points to obtain a more stable gradient, which requires only 10 outer loops and 20 inner loops for the computation. Here we also compared the time consumed for each batch of images between our method and RAP. The experiment was conducted on RTX 2080 Ti with a CUDA environment. We chose Inc-v3 as the source model and set the batch size to 10. Experimental results show that RAP will consume $207.19s$ in each batch size, while our method only consumes $29.85s$, which validates that our PGN is more computationally efficient.

## E.2    Compared with Gradient Norm Penalty

Zhao *et al.* [66] proposed a gradient norm penalty method in deep learning to improve generalization, which is similar to our method. However, there are also some differences between these two works, which are summarized as follows:

**Different goals.** Zhao *et al.* primarily focus on computing flat local minima to improve generalization during model training. In contrast, our main objective is to investigate the potential impact of flat

local minima on the transferability of adversarial attacks. Previous research in the field of adversarial attacks has shown limited attention to the relationship between flat local optima and adversarial transferability. Consequently, our work significantly contributes a new insight into the domain of adversarial attacks.

**Objective function.** A distinguishing feature of our proposed method lies in its emphasis on the gradient information surrounding the adversarial example. Specifically, we approximate the second-order Hessian matrix using the finite difference method, which plays a crucial role in our approach. The work presented by Zhao *et al*. also supports and provides theoretical foundations for the feasibility of our method within the realm of adversarial attacks.

In summary, our main motivation in this work is to explore whether flat local optimum can improve adversarial transferability. To the best of our knowledge, it is also the first work that penalized the gradient norm and the finite difference method support our motivation in the field of adversarial attacks.

# F  Additional Experimental Results

## F.1  Evaluated on Transformer-based Models

To further validate the effectiveness of our PGN method, we evaluated the performance on the Transformer-based models, *i.e.* ViT [12], PiT [19], Visformer [7], and Swin [33]. The adversarial examples are generated on Inc-v3 and the attack success rates are summarized in Table 8. It can be seen that our PGN can consistently outperform the baselines on these Transformers, showing its high effectiveness and generality to various architectures.

Table 8: Untargeted attack success rate (%) of adversarial examples in Transformer-based models. The adversarial examples are generated on Inc-v3.

| Attack | ViT | PiT | Visformer | Swin | Avg. |
|--------|------|------|-----------|------|-------|
| MI | 18.9 | 18.1 | 23.7 | 22.2 | 20.73 |
| NI | 20.0 | 19.9 | 25.5 | 25.5 | 22.73 |
| VMI | 28.4 | 34.0 | 41.1 | 40.8 | 36.08 |
| EMI | 26.0 | 27.3 | 36.9 | 36.7 | 31.73 |
| RAP | 35.2 | 41.8 | 50.6 | 50.2 | 44.45 |
| **PGN** | **44.7** | **53.2** | **65.5** | **64.5** | **56.98** |

## F.2  Attack Defense Models

In this subsection, besides normally trained models and adversarially trained models, we further validate the effectiveness of our methods on other defenses, including Bit-Red [62], ComDefend [22], JPEG [17], HGD [29], R&P [60], NIPs-r3 [39], FD [34], NPR [39], and RS [9]. The adversarial examples are generated on an ensemble of Inc-v3, Inc-v4, and IncRes-v2, and the weight for each model is $1/3$.

The experimental results are reported in Table 9. In the context of ensemble models, it is evident that our algorithm can considerably enhance existing attack methods. For instance, VMI, EMI, and RAP achieve average success rates of $54.91\%$, $61.59\%$, and $69.94\%$, respectively, against the six defense models. In contrast, our proposed PGN method achieves an average success rate of $77.08\%$, surpassing them by $22.17\%$, $15.49\%$, and $7.14\%$, respectively. This notable improvement demonstrates the remarkable effectiveness of our proposed method against adversarially trained models as well as other defense models. Consequently, it poses a more substantial threat to advanced defense models. These findings further validate that the discovery of adversarial examples within flat regions can significantly enhance the transferability of adversarial attacks.

## F.3  Attack Success Rates on CIFAR-10

To further illustrate the effectiveness of our PGN method on different datasets, we conduct experiments on CIFAR-10 [25]. we set the hyperparameters as follows: maximum perturbation $\epsilon = 8/255$, number

Table 9: Untargeted attack success rates (%) on six defense models. The adversarial examples are crafted on the ensemble models, *i.e.* Inc-v3, Inc-v4 and IncRes-v2.

| Attack | ComDefend | JPEG | NIPs-r3 | FD | R&P | HGD | Bit-Red | NPR | RS | AVG. |
|--------|-----------|------|---------|------|------|------|---------|------|------|-------|
| MI | 54.9 | 49.5 | 29.9 | 51.7 | 22.2 | 24.8 | 23.8 | 36.5 | 30.3 | 39.96 |
| NI | 56.9 | 50.8 | 29.3 | 53.6 | 23.1 | 22.3 | 23.9 | 38.1 | 80.7 | 36.52 |
| VMI | 72.4 | 71.5 | 58.1 | 67.8 | 50.6 | 54.3 | 39.0 | 44.9 | 35.6 | 54.91 |
| EMI | 78.2 | 74.1 | 69.1 | 74.8 | 60.1 | 64.8 | 47.6 | 46.8 | 38.8 | 61.59 |
| RAP | 89.5 | 88.1 | 81.0 | 79.6 | 73.1 | 72.3 | 54.6 | 49.6 | 41.7 | 69.94 |
| **PGN** | **93.7** | **91.3** | **88.3** | **85.7** | **83.6** | **82.5** | **72.1** | **51.3** | **45.2** | **77.08** |

of iterations $T = 10$, and step size $\alpha = 1/255$. We compare our PGN method with various gradient-based attacks, including MI-FGSM, NI-FGSM, VMI-FGSM, EMI-FGSM, and RAP. The adversarial examples are generated on the VGG-16, ResNet-50, and DenseNet-121 models, respectively. The results in Table 10 clearly show that our PGN method can enhance the attack transferability on the CIFAR-10 dataset. This verifies our motivation that adversarial examples located in flat local regions tend to exhibit better transferability across diverse models. Moreover, our attack method shows superior performance when applied to other datasets, reinforcing its versatility and effectiveness.

Table 10: Untargeted attack success rates (%) on the CIFAR-10 dataset for the attack methods in the single model setting. The adversarial examples are crafted on VGG-16, ResNet-50 (Res-50), and DenseNet-121, respectively.

| Attack | MobileNet | VGG-19 | GoogLeNet | Inc-v3 | DenseNet-121 | DenseNet-169 | Res-34 | Res-50 |
|--------|-----------|--------|-----------|--------|--------------|--------------|--------|--------|
| MI | 52.18 | 57.56 | 47.29 | 52.74 | 40.96 | 42.40 | 41.72 | 41.93 |
| NI | 56.13 | 61.36 | 49.19 | 54.87 | 37.61 | 39.74 | 38.74 | 38.90 |
| VMI | 66.14 | 68.05 | 60.89 | 65.63 | 55.62 | 57.21 | 55.35 | 56.46 |
| EMI | 70.69 | 74.36 | 66.78 | 70.56 | 59.98 | 63.04 | 60.47 | 61.83 |
| RAP | 77.98 | 78.43 | 73.41 | 77.86 | 68.30 | 69.74 | 65.48 | 66.27 |
| **PGN** | **85.97** | **86.73** | **82.82** | **85.59** | **72.48** | **74.66** | **71.62** | **72.95** |

(a) Untargeted attack success rates (%) for the adversarial examples crafted on VGG-16.

| Attack | MobileNet | VGG16 | VGG19 | GoogLeNet | Inc-v3 | DenseNet-121 | DenseNet-169 | Res-34 |
|--------|-----------|-------|-------|-----------|--------|--------------|--------------|--------|
| MI | 70.42 | 67.37 | 65.8 | 63.06 | 69.02 | 72.39 | 73.34 | 67.78 |
| NI | 71.97 | 65.57 | 63.76 | 63.28 | 69.13 | 71.03 | 72.78 | 65.02 |
| VMI | 77.60 | 74.27 | 73.26 | 71.11 | 75.99 | 76.80 | 77.68 | 73.54 |
| EMI | 80.11 | 78.66 | 77.43 | 76.34 | 78.12 | 79.68 | 80.12 | 77.24 |
| RAP | 86.92 | 84.24 | 83.46 | 80.68 | 81.75 | 83.54 | 84.98 | 81.36 |
| **PGN** | **90.88** | **88.68** | **88.07** | **85.79** | **89.53** | **89.93** | **90.91** | **87.19** |

(b) Untargeted attack success rates (%) for the adversarial examples crafted on Res-50.

| Attack | MobileNet | VGG16 | VGG19 | GoogLeNet | Inc-v3 | DenseNet-169 | Res-34 | Res-50 |
|--------|-----------|-------|-------|-----------|--------|--------------|--------|--------|
| MI | 63.47 | 60.64 | 60.08 | 57.39 | 63.35 | 71.09 | 61.99 | 67.37 |
| NI | 66.86 | 61.80 | 60.85 | 60.30 | 66.54 | 75.92 | 61.57 | 69.25 |
| VMI | 71.28 | 68.49 | 68.01 | 65.40 | 70.62 | 75.51 | 68.42 | 72.50 |
| EMI | 74.36 | 75.66 | 73.54 | 70.41 | 75.23 | 78.94 | 73.64 | 77.45 |
| RAP | 79.98 | 80.22 | 78.68 | 76.39 | 80.55 | 84.25 | 78.65 | 80.03 |
| **PGN** | **86.73** | **85.12** | **84.66** | **81.82** | **86.26** | **88.35** | **83.30** | **86.21** |

(c) Untargeted attack success rates (%) for the adversarial examples crafted on DenseNet-121.

