# OpenReview forum: "Boosting Adversarial Transferability by Achieving Flat Local Maxima"
_NeurIPS.cc/2023/Conference — NeurIPS 2023 poster_

### Official Review · Reviewer_2kUx · 2023-07-02

**Soundness:** 2 fair
**Presentation:** 3 good
**Contribution:** 2 fair
**Rating:** 5
**Confidence:** 4

**Summary:**

The authors empirically observed that the adversarial data lies on the flat local maxima yields enhanced attack transferability. Inspired by this observation, this paper proposes a regularization to help the gradient-based attacks to find the adversarial data at the flat local maxima. This regularization penalizes the gradient norm around the adversarial data and can be efficiently computed via the finite difference method. The empirical results validate the effectiveness of the proposed method in improving attack transferability.

**Strengths:**

1. The motivation of the proposed method is clear. The observation in Figure 1 is interesting and inspiring.

2. The authors utilized the finite difference method to make the computation more efficient.

3. It seems that the proposed method can significantly improve attack transferability.



**Weaknesses:**

1. The proposed method is not theoretically motivated, which degrades its soundness. The authors claimed the optimization of perturbation equals the model training process using an analogy. However, it lacks supportive theoretical results to support this claim. Since the aforementioned claim is not solidly proven, the reason for the adversarial data at flat local maxima yielding better transferability seems unclear.

2. The paper does not provide the standard variance of the reported results to validate their significance.

3. The paper lacks some empirical and theoretical analyses of the effectiveness of the finite difference method in speeding up optimization. I think the optimization of the gradient norm is very important for the proposed method. Therefore, the effect of the acceleration method is worth studying.


**Questions:**

1. Could the authors provide the error bar?

2. Could the authors provide a comparison between the proposed method and the previous work [1]?

3. Could the authors discuss how the effectiveness of the finite difference method in speeding up optimization empirically/theoretically? What are the computational consumption and the attack success rate with/without the finite difference method?

4. What is the performance of the proposed method compared the baseline methods evaluated on the transformer-based models (i.e., vision transformers)?

[1] Skip Connections Matter: On the Transferability of Adversarial Examples Generated with ResNets, ICLR 2020.


**Limitations:**

The motivation of the proposed method lacks theoretical support.

---

> ### Author Rebuttal · Authors · 2023-08-10
>
> Dear Reviewer 2kUx,
> Thanks for your valuable and insightful comments. We address your concerns as follows:
>
> **Q.1.** Could the authors provide the error bar?
> > **A.1.** Following your suggestions, we will add all error bars to the final version of the paper. Here we provide part of the error bars as follows. From the results, it can be seen that the standard deviation of our PGN method is not very large, which indicates that our method is stable.
> >
> > |Source:Inc-v3|Inc-v4|IncRes-v2|Res-101|
> > |:-:|:-:|:-:|:-:|
> > |VMI|74.81$\pm$0.57|70.22$\pm$0.69|75.84$\pm$0.47|
> > |EMI|81.12$\pm$0.47|76.90$\pm$0.72|80.43$\pm$0.63|
> > |RAP|82.44$\pm$0.51|79.27$\pm$0.64|81.43$\pm$0.72|
> > |PGN |91.61$\pm$0.73|90.62$\pm$0.94|89.14$\pm$0.68|
>
> **Q.2.** Could the authors provide a comparison ... and the previous work [R1]?
> > **A.2.** As you suggested, we compare our PGN with SGM [R1] using Res-18 model with MI-FGSM as the backbone method. Since SGM modifies the backpropagation and is compatible with our PGN, we also integrate PGN into SGM to evaluate its generality to other attacks. As shown in the following table, our PGN consistently exhibits better attack performance than SGM, which further shows its superiority in boosting adversarial transferability. Besides, PGN+SGM outperforms PGN as well as SGM, showing its remarkable compatibility with various attacks. We will add it in the revision.
> >
> > |Source:Res-18|Res-101|Res-152|Inc-v3|
> > |:-:|:-:|:-:|:-:|
> > |MI|82.8|73.3|54.5|
> > |SGM|89.1|82.5|66.0|
> > |PGN|95.9|92.9|77.9|
> > |PGN+SGM|96.7|93.8|80.1|
> >
> >[R1] Wu et al. "Skip Connections Matter: On the Transferability of Adversarial Examples Generated with ResNets." ICLR 2020.
>
> **Q.3.** Could the authors discuss ... with/without the finite difference method?
> >**A.3.** As you suggested, we first theoretically analyze the acceleration effect of the finite difference (FD) method. For the baseline attack method I-FGSM, the gradient is computed only once per iteration. Thus, its computational complexity is $O(n)$, where $n$ represents the image size. However, upon introducing the penalty gradient term, the need arises to compute the second-order Hessian matrix, leading to a theoretically computational complexity of $O(n^2)$.  To address this, we use the finite difference method as an approximation to the Hessian matrix, which requires the computation of the gradient twice in each iteration, effectively yielding a computational complexity of $O(2n)$. This theoretically promises significant improvements in computational efficiency.
> >
> > Additionally, we substantiate our theoretical analysis with comparative experiments. These experiments were conducted on an RTX 2080 Ti with a CUDA environment. We employed I-FGSM and evaluated the total running time on 1000 images (excluding data loading time) and the attack success rate on black-box models. The results are presented in the following table. Directly optimizing Eq. 4 results in better attack performance with high computational resources. With the finite difference method, we can better approximate the performance of direct optimization of the second-order Hessian matrices, which significantly reduces the running time and the computational memory. Furthermore, owing to the relatively modest image size ($299\times 299\times 3$) and the comparatively small number of parameters compared to the model, the accelerated computing capabilities of CUDA enable the actual running time to surpass the theoretical estimates. We will add this discussion in the revision.
> > |w/o or w/ FD|Inc-v4|IncRes-v2|Res-101| Times (Total)| Memory Size|
> > |:-:|:-:|:-:|:-:|:-:|:-:|
> > |I-FGSM (Backbone)|27.8|19.1|38.1|52 s|1631 MiB|
> > |Hessian matrix+w/o (FD)|39.2|30.2|47.0|469 s|7887 MiB|
> > |Hessian matrix+w/ (FD)|37.9|28.6|45.7|96 s|1631 MiB|
>
> **Q.4.** What is the performance ... the transformer-based models?
> >**A.4.** Please refer to the response to Q.1 of Reviewer PZXp.
>
> **Q.5.** The proposed method is not ... transferability seems unclear.
> >**A.5.** In [R1], Lin et al. analogize the adversarial example generation process to the standard neural model training process, where the input $x$ can be viewed as parameters to be trained and the target model can be treated as the training set. From this prespective, the transferability of adversarial examples is equivalent to the generalization of the normally trained models. Intuitively, training model and adversarial perturbation generation are both optimization problems, which derives such analogy. Besides, this analogy has been widely accepted by nemerous works, such as exploring better optimization methods (NI [R1], VMI [R2]) or data augmentation methods (ODI [R3], DITL [R4]), which are effective in improving the transferability of adversarial examples.
> >
> > In fact, it is difficult to provide a theory to support the analogy between adversarial transferability and model generalization in the field of adversarial attacks. In this work, we are inspired by this analogy and attempted to enhance the transferability of adversarial examples from a new perspective. Hence, we try to explore flat local minima to enhance the transferability of adversarial examples. We assumed and experimentally verified that the flat local optima are related to the transferability of the adversarial examples, and the loss surface maps of the generated adversarial examples in Fig. 2 also validates our motivation. Also, we will keep studying the theoretical connection between transferability and flat local minima in our future work.
> >
> > [R1] Lin et al. "Nesterov Accelerated Gradient and Scale Invariance for Adversarial Attacks." ICLR. 2020.
> >
> > [R2] Wang et al. "Enhancing the Transferability of Adversarial Attacks through Variance Tuning." CVPR. 2021.
> >
> > [R3] Byun et al. "Improving the Transferability of Targeted Adversarial Examples through Object-Based Diverse Input." CVPR. 2022.
> >
> > [R4] Yuan et al. "Adaptive Image Transformations for Transfer-based Adversarial Attack." ECCV. 2022.

---

> > ### Comment · Reviewer_2kUx · 2023-08-14
> > **Understand**
> >
> > Thanks for your response. The empirical results seem to sufficiently support the effectiveness of the proposed method. However, from the theoretical perspective, this paper does not provide a rigorous theoretical guarantee of its effectiveness. Therefore, I still would like to lean toward Boraberline Accept. I will not defend for its acceptance.

---

### Official Review · Reviewer_8Bpx · 2023-07-05

**Soundness:** 2 fair
**Presentation:** 3 good
**Contribution:** 2 fair
**Rating:** 3
**Confidence:** 4

**Summary:**

The paper proposes a method called Penalizing Gradient Norm (PGN) to improve the transferability of adversarial perturbations. The method is motivated by the observation that encouraging the flatness of the local landscape for adversarial examples can lead to better transferability, and thus PGN regulates the process of gradient-based adversarial attack algorithms by penalizing the magnitude of the loss gradient with respect to the input. Since such a regularization process requires the input Hessian which is a computationally expensive process, PGN utilizes the finite difference method to approximate the Hessian matrix. Experiments on the ImageNet-compatible dataset demonstrate that the proposed method can improve the transferability of untargeted attacks in comparison to other baseline methods.

**Strengths:**

Originality: The proposed method is original and intuitive.

Clarity: The general structure of the paper is very clear: moving from validating an assumption to proposing an algorithm, and finally evaluating the proposed method with empirical results.

Significance: The proposed method addresses a practical security concern of deep learning models. The proposed method improves the transferability of the adversarial perturbations compared to existing gradient-based methods. Extensive empirical evaluations were performed to demonstrate the efficacy of the proposed method.

**Weaknesses:**

Reverse adversarial perturbation (RAP) is a closely related work that encourages adversarial examples to be located at a region with low loss values. To demonstrate the novelty and significance of the proposed method, the paper needs a detailed discussion of the differences and similarities compared to RAP.

One of the major contributions claimed by the paper is the empirical validation that "adversarial examples located in flat regions have good transferability", and it is mainly covered in Sec. 3.2.
Putting aside the significance of the contribution, the authors should be very careful about the claims and statements in Sec 3.2. The assumption and the followed empirical validation both suffer from the lack of rigor and thus weaken the significance of the contributions. Please see the Questions section for additional discussions.

Some technical details in Sec 3.3 require clarifications.

**Questions:**

The issue around Sec. 3.2 stems from the lack of rigor in Assumption 1.
How is a local region defined? What is the definition of flatness, and how to measure it?
Following the assumption, why is the maximum l2 norm used in (3)? not a value averaged over l2 norm of data points sampled around x?
Consider inputs that fail to transfer, but now are transferable because of the modified objective. Are they indeed situated in a flat region?
More importantly, why is (3) necessary to validate the assumption? Since the assumption is agnostic to the attack algorithm, it should be true to any adversarial attack methods: I-FGSM, MI-FSGM, VIM-FGSM, etc. As such, given any attack algorithm, we should compare the flatness between adversarial examples that are transferable and those which fail to transfer.

Ln 163 requires clarification, why do we have such expectations?

The FD method circumvents the expensive computation of the input Hessian, and the approximation becomes accurate with decreasing value of \alpha. Why is the \alpha used in FD the same as the \alpha used in the iterative process of generating adversarial examples? To achieve an accurate approximation of the Hessian, shouldn't the stepsize used in FD be very small? \alpha = \eps/T seems to be a large value to me.

With the current choice of \alpha, how close is the approximation (5) to the actual Hessian?

Evaluation:
Is PGN based on standard ifgsm, or one of the momentum variants? Since RAP is the closes method to PGN, is RAP used in the evaluation based on the standard ifgsm as well?

Ln303: The statement of "flat local minima result in better generalization" being a fact is quite strong.

Also, I would suggest the author proofread the paper. There are several very noticeable typos. For instance, even the name of the method is spelled incorrectly as "Penalizing" (Ln48, Ln55)

**Limitations:**

I suggest the author include a brief discussion of the limitation of the proposed work.

---

> ### Author Rebuttal · Authors · 2023-08-10
>
>
> Dear Reviewer 8Bpx,
> Thanks for your valuable and insightful comments. We address your concerns as follows:
>
> **Q.1.** PGN v.s. RAP
> >**A.1.** As you suggested, we discuss the similarities and differences between PGN and RAP methods as follows.
> >1) **Empirical verification**. Both RAP and PGN aim to achieve flat local optima for more transferable adversarial examples. Besides, we are the first work that provides an empirical verification that adversarial examples in flat local optima are more transferable.
> >2) **Methodology**. RAP injects worst-case perturbations into the optimization process to maximize the loss and achieve falt local optima. In contrast, PGN introduces a gradient norm into the loss to achieve a flat local optimum. Hence, the method we employed differs significantly from RAP.
> >3) **Efficiency**. Our method is more computationally efficient. In RAP the outer loop is 400 and the inner loop is 8. In contrast, our PGN method employs finite differences to approximate the Hessian matrix and samples multiple points to obtain a more stable gradient, which requires only 10 outer loops and 20 inner loops for the computation. Thus, our method not only improves computational efficiency but also achieves a higher attack success rate.
>
> **Q.2.** The issue around Sec. 3.2...which fail to transfer.
> >**A.2.**
> >1) In mathematics, we define the example $x'$ in the $\epsilon$-neighborhood of the input image $x$ as a local region. Flatness indicates the maximum gradient value in the neighborhood of the sample. The smaller the gradient value is, the flatter it is. To characterize the flatness of the loss surface, we define the slope $k=\frac{J(x_0)-J(x_i)}{\||x_0 - x_i \||_2}$, where $x_0$ represents the center point of the 2D map in Fig. 2, and $x_i$ represents the points sampled near the center point $x_0$.
> >2) During the optimization process, there might be saddle points. If we use averaged gradient, saddle points will reduce the size of the averaged gradient value, making us unable to perceive the sharper regions. Hence we use the maximum gradient, which can better perceive the worst-case gradient.
> >3) To address your concerns, we selected images (about 286 images) that can be transferred using our PGN method but cannot be transferred using MI-FGSM. We calculate the average slope $k$ of adversarial examples on the Inc-v3 model for these 286 images. **See the first table in PDF**. The adversarial example generated by our PGN have smaller slopes, confirming that our method does craft adversarial examples in flatter local regions.
> >4) The goal of adding the penalized gradient norm is to make the gradient value smaller, and a smaller gradient value also means that the local region is flatter. Thus, if we verify that Eq.3 can improve the adversarial transferability, it naturally verifies that our assumption is valid, i.e., flat local optima have better transferability.
> >5) Our assumption is agnostic to the attack algorithm. Hence, PGN is suitable for any gradient-based attack methods. To address your concerns, we combine PGN with these gradient-based attack methods to generate adversarial examples on Inc-v3 and compute their flatness (averaged on 1000 images). **See the second table in PDF**. Our PGN method can effectively improve the adversarial transferability of existing attacks and make the adversarial examples located in flatter regions.
>
> **Q.3.** Ln 163 requires...expectations?
> >**A.3.** Intutitively, it is expected that the loss of these data points in a small neighboorhood are similar. To address your concerns, we evaluated the value of $J(x_t^{adv})$ and $J(x')$ during the iteration process. In each iteration, we first calculated the value of the loss function for the adversarial example $x_t^{adv}$, then we randomly sampled a sample $x'$ in the neighborhood of the $x_t^{adv}$ and calculated the loss function $J(x')$. **See the third table in PDF**. It can be observed that both loss function values are relatively approximate. We can also observe that the expectation of $J(x')$ is smaller than that of $J(x^{adv})$, and maximizing a smaller value during the optimization process is more beneficial for our optimization.
>
> **Q.4.** $\alpha=\epsilon/T$ seems to be a large value
> >**A.4.** We conduct PGN using smaller step sizes **see the fourth table in PDF**. We can see smaller step sizes cannot improve adversarial transferability. This is because the smaller the step size, the closer two neighboring gradients are, leading our method to degrade to MIFGSM. To avoid redudant hyper-parameters, we simply adopt the step size of $\alpha$ in this paper.
>
> **Q.5.** With the current...the actual Hessian?
> >**A.5.** To address your concerns, we compared the cosine similarity of perturbations generated by the Hessian matrix and finite differences method. The average cosine similarity of 1000 images is about 0.9035. It can be observed that the current finite difference can effectively approximate the adversarial perturbation generated by the Hessian matrix.
>
> **Q.6.** Evaluation: Is PGN...well?
> >**A.6.** For fair comparison, our PGN and RAP adopt MI-FGSM for evaluation.
>
> **Q.7.** Ln303: The...being a fact is quite strong.
> >**A.7.** Existing works [R1, R2, R3] have shown that more flat local minima often result in better model generalization from empirical and theoretical perspectives. We will rephrase that sentence as "Inspired by the observation that flat local minima often result in better generalization".
> >
> >[R1] Keskar et al. On large-batch training for deep learning: Generalization gap and sharp minima. ICLR. 2017.
> >
> >[R2] Neyshabur et al. Exploring generalization in deep learning. NeurIPS. 2017.
> >
> >[R3] Foret et al. Sharpness-aware minimization for efficiently improving generalization. ICLR. 2021.
>
> **Q8-Q10**
> > We have corrected the typos and a discussion of the limitations will be added in the revised paper. Technical details in Sec 3.3 have be provided in Appendix.

---

> > ### Comment · Reviewer_8Bpx · 2023-08-18
> >
> > I appreciate author's detailed response. One of my major concern with the paper is the lack of rigour, particularly in Sec 3. As such, I will maintain the initial score.

---

> > > ### Author Response · Authors · 2023-08-20
> > >
> > > Thank you for responding to our comments. In terms of your remaining concern, "The issue around Sec. 3.2 stems from the lack of rigor in Assumption 1", we will address it as follows:
> > >
> > > **1. We provide a more rigorous description of Assumption 1 here.**
> > > > **Assumption 1**: Given the maximum radius $\zeta$ for the local region and two adversarial examples $x_1^{adv}$ and $x_2^{adv}$ for the same input image $x$, if $\max _{x' \in \mathcal{B} _{\zeta}(x _1^{adv})} \| \nabla _{x'}J(x', y;\theta) \| _2 < \max _{x' \in \mathcal{B} _{\zeta}(x _2^{adv})} \| \nabla _{x'}J(x', y;\theta) \| _2$, $x _1^{adv}$ tends to be more transferable than $x _2^{adv}$ across various models.
> > > >
> > > > Here we adopt the maximum gradient in the neighborhood to evluate the flatness of local region, which is more rigorous.
> > >
> > > **2. Regarding "The assumption and the followed empirical validation both suffer from the lack of rigor and thus weaken the significance of the contributions", we need to clarify again as follows.**
> > > >In this work, inspired by the observation that flat local minima can bring better generalization during the model training process, we try to explore whether flat local optimum can improve the adversarial transferability from a new perspective. Hence, we first propose the assumption that adversarial example at a flat local region tends to have better transferability. To validate Assumption 1, we introduce a regularizer to minimize the maximum gradient in the $\epsilon$-neighborhood of the original objective loss function. Intuitively,  the smaller gradient also indicates a flatter location. By optimizing this new objective loss function (Eq.3), we find that adversarial examples have better transferability. Thus, when we verify that Eq.3 can improve the adversarial transferability, it naturally verifies that our assumption is valid, i.e., flat local optima have better transferability. Based on this assumption and verification, we propose a novel attack to boost adversarial transferability.
> > >
> > > **3. More clarifications using the finite difference methods for approximation**
> > > > In line 166, we have noted that existing adversarial attacks typically rely on the sign of the gradient, rather than requiring an exact gradient value. Thus, we approximate the second-order Hessian matrix using the finite difference method to accelerate the attack process. We also have compared the cosine similarity of perturbations generated by the Hessian matrix and finite differences method and measured an average similarity of 0.9035 for 1000 images. It can be observed that the current finite difference can effectively approximate the adversarial perturbation generated by the Hessian matrix.
> > >
> > > Thank you for your effort and reviews. We are looking forward to your further reply and happy to address your concerns if any.

---

### Official Review · Reviewer_xHRb · 2023-07-06

**Soundness:** 4 excellent
**Presentation:** 4 excellent
**Contribution:** 3 good
**Rating:** 5
**Confidence:** 5

**Summary:**

This paper aims to boost adversarial transferability by using the Penelizing Gradient Norm, which can restrict adversarial examples located in flat regions. The writing is good, and it is easy to read. The experiments demonstrate that the proposed method achieves good results.

**Strengths:**

- The motivation is clear, and the writing is well.

-  The analysis in Sec.3.2 is interesting, which can verify the assumption.

- The proposed method is simple but effective.



**Weaknesses:**

1. In Theorem 1, the authors briefly introduce the finite difference method, which is fundamental for efficiently approximating a second-order Hessian matrix. Although this is an interesting solution, it is better to verify this approximation in experiments if Eq. 6 is a good approximate solution to the objective function in Eq. 4. On one hand, I think the results of solving Eq. 4 directly should be reported, which can show that the approximated solution will not affect the performance. On the other hand, the running time and complexity analysis are also considered, which can show that this key design of PGN actually works well.

2. In Table 1, we first observe that PGN is a good solution for boosting transferability. But, we can also observe that the source model IncRes-v2 can achieve the best average score. Does this mean that the loss surface of this model is in a more smooth region?

3. I understand this paper focuses on boosting adversarial transferability. However, for black-box attacks, query-based adversarial attacks are also widely studied. Therefore, I am interested in if the Penelizing Gradient Norm is a generally method for black-box attacks, not limited to transfer attacks.



**Questions:**

See above weakness.

**Limitations:**

The proposed method is similar to the previous work [a].

[a] Penalizing gradient norm for efficiently improving generalization in deep learning. ICML 2022.

---

> ### Author Rebuttal · Authors · 2023-08-10
>
> Dear Reviewer xHRb,
> Thanks for your valuable and insightful comments. We address your concerns as follows:
>
> **Q.1.** In Theorem 1, the authors briefly ... key design of PGN actually works well.
> >**A.1.** As you suggested, we first theoretically analyze the acceleration effect of the finite difference (FD) method. For the baseline attack method I-FGSM, the gradient is computed only once per iteration. Thus, its computational complexity is $O(n)$, where $n$ represents the image size. However, upon introducing the penalty gradient term, the need arises to compute the second-order Hessian matrix, leading to a theoretically computational complexity of $O(n^2)$.  To address this, we use the finite difference method as an approximation to the Hessian matrix, which requires the computation of the gradient twice in each iteration, effectively yielding a computational complexity of $O(2n)$. This theoretically promises significant improvements in computational efficiency.
> >
> > Additionally, we substantiate our theoretical analysis with comparative experiments. These experiments were conducted on an RTX 2080 Ti with a CUDA environment. We employed I-FGSM and evaluated the total running time on 1000 images (excluding data loading time) and the attack success rate on black-box models. (**The results please refer to the response to Q.3. of Reviewer 2kUx**).  Directly optimizing Eq. 4 results in better attack performance with high computational resources. With the finite difference method, we can better approximate the performance of direct optimization of the second-order Hessian matrices, which significantly reduces the running time and the computational memory. Furthermore, owing to the relatively modest image size ($299\times 299\times 3$) and the comparatively small number of parameters compared to the model, the accelerated computing capabilities of CUDA enable the actual running time to surpass the theoretical estimates. We will add this discussion in the revision.
>
> **Q.2.** In Table 1, we first ... is in a more smooth region?
> > **A.2.** We tested the smoothness over the loss function of the adversarial examples generated by Inc-v3 and IncRes-v2 models, respectively. To characterize the flatness of the loss surface, we define the slope $k=\frac{J(x_0)-J(x_i)}{\||x_0 - x_i \||_2}$, where $x_0$ represents the center point of the 2D map in Fig. 2, and $x_i$ represents the points sampled near the center point $x_0$. Notably smaller values of $k$ mean flatter. To derive comprehensive insights, we randomly selected a subset of images ($S_i$, which denotes the $i$-th image) and uniformly sampled 10 points near the center to calculate their average slope $k$.
> >
> > |Averaged slope $k$|$S_0$|$S_1$|$S_2$|$S_3$|$S_4$|
> > |:-:|:-:|:-:|:-:|:-:|:-:|
> > |Inc-v3|0.013|0.254|0.016|0.024|0.009|
> > |IncRes-v2|0.004|0.122|0.018|0.020|0.007|
> >
> > From the table, it can be concluded that most of the adversarial examples generated by the IncRes-v2 model have smaller slopes. We also counted the slopes of 1000 images on the Inc-v3 and IncRes-v2 models, respectively. The results show that 76.4% of the images exhibit smaller slopes on the IncRes-v2 model. The observations from these experiments confirm that the IncRes-v2 model facilitates the generation of adversarial examples that exist in smoother regions of the loss landscape. In our future work, we will investigate the possible reason and design effective method to train the model with more smooth region as the surrogate model for better transferability.  Thanks for your insightful comments.
>
> **Q.3.** I understand this paper ... not limited to transfer attacks.
> > **A.3.** Thanks for this valuable comment. Unfortunately, our PGN might not generalize to query-based attack methods. Here, we provide the following analysis.
> > 1) The goal of query-based attack methods is different from ours. The goal of our method is more concerned with improving the transferability of the adversarial examples on different black-box models, while the query-based method mainly focuses on the attack success rate on the source model.
> > 2) Query-based attack methods mainly exploit limited output information such as labels and logits, while our PGN needs to utilize the gradient to the objective loss function. Hence our method is not inherently applicable on query-based attack methods.
>
> **Q.4.** The proposed method is similar to the previous work [R1].
> > **A.4.** Indeed, both our PGN and [R1] penalize the gradient norm. However, there are significant differences between these two works, which are summarized as follows:
> > 1) **Different goals**. While [R1] primarily focuses on enhancing flat local minima to improve generalization during model training, our main objective is to investigate the potential impact of flat local minima on the transferability of adversarial attacks. Previous research in the field of adversarial attacks has shown limited attention to the relationship between flat local optima and adversarial transferability. Consequently, our work significantly contributes new insights to the domain of adversarial attacks.
> > 2) **Objective function**. A distinguishing feature of our proposed method lies in its emphasis on the gradient information surrounding the adversarial example. Specifically, we approximate the second-order Hessian matrix using the finite difference method, which is crucial for our approach. The work presented in [R1] also supports and provides theoretical foundations for the feasibility of our method within the realm of adversarial attacks.
> >
> > In summary, our main motivation in this work is to explore whether flat local optimum can improve the adversarial transferability.  To the best of our knowledge, it is also the first time to use penalized the gradient norm and finite difference method to support our motivation in the field of adversarial attacks.
> >
> > [R1] Zhao et al. "Penalizing gradient norm for efficiently improving generalization in deep learning." ICML 2022.

---

> > ### Comment · Reviewer_xHRb · 2023-08-18
> > **Response to authors' rebuttal**
> >
> > I appreciate the authors' detailed response to the initial review. Having carefully considered their feedback in conjunction with the comments from other reviewers, I decided to maintain my initial rating.

---

### Official Review · Reviewer_PZXp · 2023-07-07

**Soundness:** 3 good
**Presentation:** 3 good
**Contribution:** 3 good
**Rating:** 7
**Confidence:** 4

**Summary:**

In this work, the authors first assume and empirically validate that adversarial examples at flat local minima tend to have better adversarial transferability. Based on this finding, they introduce a regularizer on the gradients in the neighborhood of the input sample to achieve flat local minima. To make the attack more computationally efficient, they propose Penalizing Gradient Norm (PGN) attack, which approximates the second-order Hessian matrix by interpolating two Jacobian matrixes.

**Strengths:**

It is the first work that empirically validates that adversarial examples at flat local minima have better adversarial transferability.

The approximation on the second-order Hessian matrix is reasonable with theoretical support.

The proposed method is simple yet effective. Extensive experiments have shown that PGN can significantly boost adversarial transferability compared with existing methods.

The visualization in Figure 2 validates that PGN can achieve better flat local minima than existing attacks, which further supports their motivation.

**Weaknesses:**

Results on vision transformers, such as ViT, Swin, etc. would better be included.

Evaluations on more defense methods, such as randomized smoothing and denoising should be conducted.

**Questions:**

In Line 9 of Algorithm 1, g’ and g* are only related to i-th sampled example x’. Is it a typo? I think it should accumulate all the gradients of N sampled examples.
What is the difference between the two symbols L and J? I think they are both loss functions. Definitions of these two loss functions should be clarified.

**Limitations:**

No.

---

> ### Author Rebuttal · Authors · 2023-08-09
>
> Dear Reviewer PZXp,
> Thanks for your valuable and insightful comments. We address your concerns as follows:
>
> **Q.1.** Results on vision transformers, such as ViT, Swin, etc. would better be included.
>
> > **A.1.** As you suggested, we further adopt four mainstream vision transformers to evaluate the effectiveness of our method, i.e., ViT [R1], PiT [R2], Visformer [R3], and Swin [R4]. The adversarial examples are generated on Inc-v3 and the attack success rates are summarized in the following table. It can be seen that our PGN can consistently outperform the baselines on these transformers, showing its high effectiveness and generality to various architectures. We will report the complete results in our final paper.
> >
> > |Method	|ViT	|PiT	|Visformer	|Swin|
> > | :----: | :----: | :----: | :----: | :----: |
> > |MI	|18.9	|18.1	|23.7	|22.2 |
> > |NI	|20.0	|19.9	|25.5	|25.5 |
> > |VMI	|28.4	|34.0	|41.1	|40.8 |
> > |EMI	|26.0	|27.3	|36.9	|36.7 |
> > |RAP	|35.2	|41.8	|50.6	|50.2 |
> > |PGN (ours)	|**44.7**	|**53.2**	|**65.5**	|**64.5** |
> >
> > [R1] Alexey et al. "An image is worth 16x16 words: Transformers for image recognition at scale." ICLR. 2021.
> >
> > [R2] Heo et al. "Rethinking spatial dimensions of vision transformers." ICCV. 2021.
> >
> > [R3] Chen et al. "Visformer: The vision-friendly transformer." ICCV. 2021.
> >
> > [R4] Liu et al. "Swin transformer: Hierarchical vision transformer using shifted windows." ICCV. 2021.
>
>
> **Q.2.** Evaluations on more defense methods, such as randomized smoothing and denoising should be conducted.
>
> > **A.2.** Following your suggestion, we further evaluate our PGN and other gradient-based attacks on three advanced defense models, i.e., Feature Distillation (FD) [R1], Randomized Smoothing (RS) [R2], and Neural Representation Purifier (NRP) [R3]. The adversarial examples are generated on the ensemble models, i.e. Inc-v3, Inc-v4 and IncRes-v2, and the attack success rates are summarized in the following table. As we can see, our PGN can consistently exhibits better attack performance than the baselines on these advanced defenses. These results futher validate the superiority of our proposed PGN. We will add the complete results in our final paper.
> >
> > | Method | FD | RS | NPR|
> > | :----: | :----: | :----: | :----: |
> > | MI | 51.7 | 30.3 | 36.5 |
> > | NI | 53.6 | 30.7 | 38.1 |
> > | VMI| 67.8 | 35.6 | 44.9 |
> > | EMI| 74.8 | 38.8 | 46.8 |
> > | RAP| 79.6 | 41.7 | 49.6 |
> > | PGN (ours)| **85.7** | **45.2** | **51.3** |
> >
> > [R1] Liu et al. "Feature Distillation: DNN-Oriented JPEG Compression Against Adversarial Examples." CVPR. 2019.
> >
> > [R2] Cohen et al. "Certified adversarial robustness via randomized smoothing." ICML. 2019.
> >
> > [R3] Naseer et al. "A self-supervised approach for adversarial robustness." CVPR. 2020.
>
> **Q.3.** In Line 9 of Algorithm 1, g’ and g* are only related to i-th sampled example x’. Is it a typo? I think it should accumulate all the gradients of N sampled examples. What is the difference between the two symbols L and J? I think they are both loss functions. Definitions of these two loss functions should be clarified.
> > **A.3.** Thanks for pointing out this typo.
> > 1) In fact, the symbol $\bar{g}$ is intended to accumulate the gradients of multiple sampled examples, and the correct representation should be $\bar{g} = \bar{g}+(1-\delta)\cdot g'+ \delta \cdot g^{\ast}$. Specifically, in each iteration, $\bar{g}$ is initialized to $0$. Then $N$ different examples $x'$ will be randomly sampled in the neighborhood of $x_t^{adv}$. Subsequently, the gradients $g'$ and $g^{\ast}$ related to example $x'$ will be computed. Finally, these gradients will be accumulated into $\bar{g}$ as the final gradient. We will correct this typo in Algorithm 1 in the revised version.
> >
> > 2) The symbols $L$ and $J$ represent two different loss functions. Where $J$ is the original loss function of the classifier $f$ (e.g., the cross-entropy loss), and $L$ is our proposed loss function that introduces a penalized gradient norm to the original loss function to achieve flat local maxima. To avoid confusion, we will add the above clarifications to the final version of the paper.
>
> We appreciate your efforts in improving the clarity and accuracy of our paper, and we will make the necessary revisions to address these issues.

---

> > ### Comment · Reviewer_PZXp · 2023-08-20
> >
> > I have carefully read the responses and the other reviews. I think the authors have addressed my concerns. Also, the assumption is reasonable with empirical verfication, which makes the proposed method novel and solid. Thus, I raise my score to 7.

---

### Author Rebuttal · Authors · 2023-08-10

Dear Reviewers and Area Chairs:

We sincerely appreciate all of your precious time and constructive comments.  The results of the partial response data (Tables) are in the pdf file.

---

### Decision · Program_Chairs · 2023-09-21

**Decision:**

Accept (poster)

**Comment:**

Overall, this is a borderline paper.  The reviewer 8Bpx gave 3 score and provided many valuable comments. AC read the discussion between authors and Reviewer 8Bpx. The main concern of this paper is the Sec 3.2 is not rigorous. AC agrees with the reviewer's comment by reading the paper. However, when reading the rebuttal, AC feels that the authors provide good explanation. Although this paper lacks of  supportive theoretical results, overall, this paper provides new insights and understanding in term of transferability of adversarial examples. AC tends to accept this paper. However, AC indeed viewed the reviewers' comments are very valuable and the authors should add them in the revised version.